# SEEDVR2: ONE-STEP VIDEO RESTORATION VIA DIFFUSION ADVERSARIAL POST-TRAINING

**Jianyi Wang**[1,2*♣]    **Shanchuan Lin**[2]    **Zhijie Lin**[2]    **Yuxi Ren**[2]    **Meng Wei**[2]
**Zongsheng Yue**[1]    **Shangchen Zhou**[1]    **Hao Chen**[2]    **Yang Zhao**[2]    **Ceyuan Yang**[2]
**Xuefeng Xiao**[2]    **Chen Change Loy**[1]    **Lu Jiang**[2♣]

[1] Nanyang Technological University          [2] ByteDance Seed
https://iceclear.github.io/projects/seedvr2/

## ABSTRACT

Recent advances in diffusion-based video restoration (VR) demonstrate significant improvement in visual quality, yet yield a prohibitive computational cost during inference. While several distillation-based approaches have exhibited the potential of one-step image restoration, extending existing approaches to VR remains challenging and underexplored, particularly when dealing with high-resolution video in real-world settings. In this work, we propose a one-step diffusion-based VR model, termed as SeedVR2, which performs adversarial VR training against real data. To handle the challenging high-resolution VR within a single step, we introduce several enhancements to both model architecture and training procedures. Specifically, an adaptive window attention mechanism is proposed, where the window size is dynamically adjusted to fit the output resolutions, avoiding window inconsistency observed under high-resolution VR using window attention with a predefined window size. To stabilize and improve the adversarial post-training towards VR, we further verify the effectiveness of a series of losses, including a proposed feature matching loss without significantly sacrificing training efficiency. Extensive experiments show that SeedVR2 can achieve comparable or even better performance compared with existing VR approaches in a single step.

## 1 INTRODUCTION

Diffusion models (Liu et al., 2023; Ho et al., 2020; Rombach et al., 2022; Song et al., 2021) are becoming the the de-facto model for real-world image restoration (IR) (Wang et al., 2024a; Yu et al., 2024; Lin et al., 2024a; Yue et al., 2023; 2025; 2024) and video restoration (VR) (Wang et al., 2025; Zhou et al., 2024; Yang et al., 2024; Xie et al., 2025; Li et al., 2025a). Though these approaches show promise in generating realistic details, they typically require tens of steps to generate a video sample, leading to considerably high computational cost and latency. Such significant cost is further amplified when processing long videos at high resolutions.

Inspired by recent advances in diffusion acceleration (Sauer et al., 2024a; Yin et al., 2024a; Luo et al., 2025; 2023a), several one-step diffusion IR approaches (Yue et al., 2025; Wang et al., 2024b; Zhu et al., 2024; Li et al., 2025b; Dong et al., 2025; Xie et al., 2024; Li et al., 2024; Wu et al., 2024) have been proposed, showing potential in generating promising results comparable to that of multi-step approaches. The majority of these methods (Wang et al., 2024b; Zhu et al., 2024; Li et al., 2025b; Dong et al., 2025; Xie et al., 2024; Wu et al., 2024) rely on distillation from a pre-trained teacher model, suffering from an undesired upper bound constrained by the teacher model. The high computational cost of the teacher model further makes it less practical to apply these methods to VR. The closest to our work are recent distillation-free one-step IR methods that either learn from a discriminator prior (Li et al., 2024) or a generative prior (Yue et al., 2025; Zhang et al., 2024). These methods save computational cost by training on an implicit teacher model, *i.e.*, diffusion prior (Sauer et al., 2024b; Podell et al., 2024) with LoRA (Hu et al., 2022a). Given the limited capability of

---

[*]Work was done during Jianyi Wang's internship at ByteDance Seed (iceclearwjy@gmail.com).
[♣] Now at Apple.

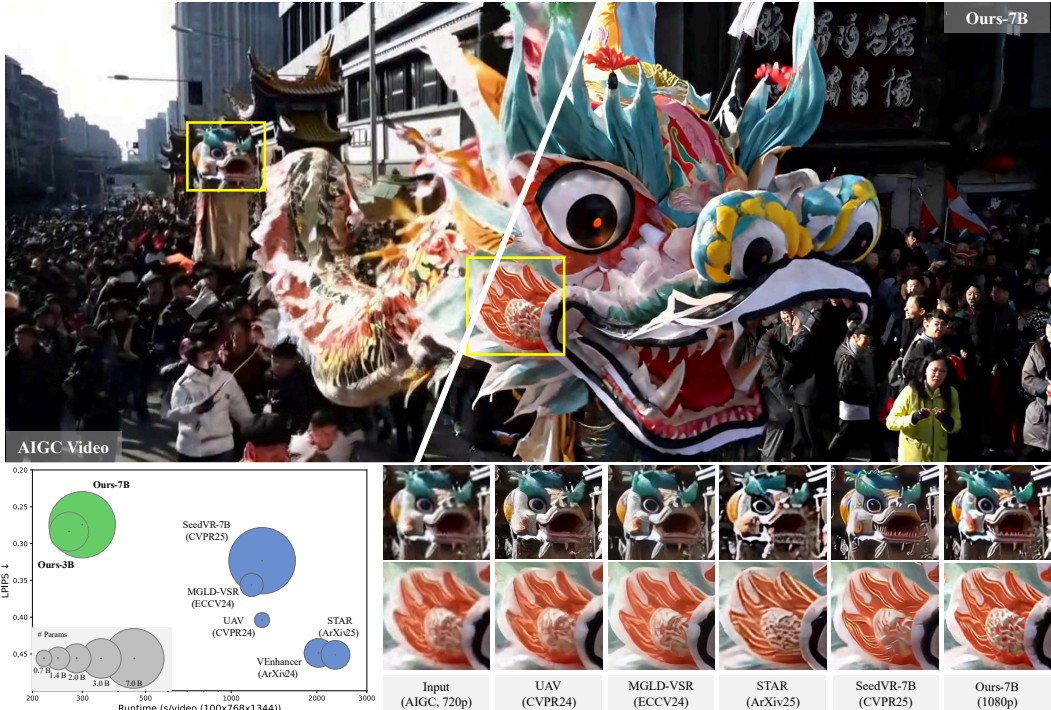

Figure 1: Speed and performance comparisons. Our SeedVR2 demonstrates impressive restoration capabilities, offering fine details and enhanced visual realism. While achieving comparable performance with SeedVR (Wang et al., 2025), our SeedVR2 is over $4\times$ faster than existing diffusion-based video restoration approaches (Zhou et al., 2024; Yang et al., 2024; He et al., 2024a; Xie et al., 2025) (We use 50 sampling steps for these baselines to maintain stable performance), even with at least four times the parameter count (**Zoom-in for best view**).

existing video diffusion as prior, our work turns to explore one-step video restoration via adversarial training without frozen priors, making it possible to alleviate the bias learned by these models.

Achieving one-step VR, especially under high resolutions, is challenging, yet underexplored. In this paper, we introduce a new method, SeedVR2, for one-step VR towards real-world scenarios. Our method follows Adversarial Post-Training (APT) (Lin et al., 2025) to adopt a pre-trained diffusion transformer, *i.e.*, SeedVR (Wang et al., 2025) as initialization, and continues to fully tune the whole network using the adversarial training objective against real data. Compared with previous one-step IR methods, SeedVR2 eliminates the substantial cost associated with pre-computing video samples from the diffusion teacher during distillation. Moreover, without the constraint from a diffusion teacher or prior, SeedVR2 presents the potential to surpass the initial model, demonstrating comparable or even superior performance to multi-step VR diffusion models.

While it is applicable to directly adopt APT for VR, we empirically observe several key aspects that can be improved based on the nature of VR. **First**, given the low-quality input as a condition, we observe a more stable training process of VR compared with text-to-video generation (Lin et al., 2025), *i.e.*, no obvious mode collapse is observed with only a single stage of adversarial training. However, we notice a performance drop when handling heavy degradations. We hereby adopt a progressive distillation (Salimans and Ho, 2022) before the adversarial training to maintain the restoration capability under one-step generation. **Second**, when applying window attention with a predefined window size on high-resolution VR, *e.g.*, over 2K resolution, we observe visible boundary artifacts between window patches. We conjecture this is due to the improper settings of the window size and training resolutions, *e.g.*, too large window sizes compared with relatively small training resolutions, making the model insufficiently trained on handling window shifting. Such a predefined window manner may further limit the robustness of 3D Rotary Positional Embedding (RoPE) (Su et al., 2024) inside each window when dealing with inputs with various resolutions. To tackle this problem, we propose an adaptive window attention mechanism to dynamically adjust the window size within a certain range, significantly improving the robustness of the model when handling

arbitrary-resolution inputs. **Third**, adversarial training with the exceptionally large generator and discriminator can be unstable even with APT, *i.e.*, a performance deterioration can be observed after long training, *e.g.*, 20k iterations. We follow Huang et al. (2024) to enhance the training stability by introducing RpGAN (Jolicoeur-Martineau, 2019) and an additional approximated R2 regularization loss. While L1 loss and LPIPS loss (Zhang et al., 2018a) are commonly used in VR training for better perception-distortion tradeoff (Blau and Michaeli, 2018), the necessity to calculate LPIPS in pixel space makes it unaffordable for high-resolution video training. Training a latent LPIPS model (Kang et al., 2024a) is also not applicable due to the lack of video-specific data. We instead propose a feature matching loss to replace the LPIPS loss for efficient adversarial training. Specifically, we directly extract multiple features from different layers of the discriminator and measure the feature distance between the prediction and ground-truth. We empirically show that such a feature matching loss is an effective alternative in our case.

To our knowledge, SeedVR2 is among the early attempts to demonstrate the feasibility of one-step video restoration or super-resolution using a diffusion transformer. Benefiting from the adversarial training with specific designs for VR, we are able to train the largest-ever VR GAN ($\sim$16B for the generator and discriminator in total), which can achieve high-quality restoration in a single sampling step with high efficiency. The main contributions of our work are as follows:

- We present an effective adaptive window attention mechanism, enabling efficient high-resolution (*e.g.*, 1080p) restoration in a single forward step with faithful details, as shown in Figure 1.
- With the adversarial post-training framework, we explore effective design improvements specific to video restoration, focusing on the loss function and progressive distillation.

Extensive experiments validate the effectiveness of our design, and demonstrate the superiority of our method over existing methods, both quantitatively and qualitatively.

## 2 RELATED WORK

**Video Restoration.** Traditional video restoration (VR) methods (Chan et al., 2021; 2022a; Liang et al., 2024; 2022; Li et al., 2023; Chen et al., 2024; Youk et al., 2024; Wang et al., 2019) primarily concentrate on synthetic datasets, suffering from limited effectiveness in real-world scenarios. More recent efforts (Chan et al., 2022b; Xie et al., 2023; Zhang and Yao, 2024) have shifted focus towards real-world scenarios, but still struggle with generating realistic textures due to constrained generative capabilities. Inspired by the rapid progress in diffusion models (Ho et al., 2020; Rombach et al., 2022; Sohl-Dickstein et al., 2015; Yang et al., 2021; Nichol et al., 2022; Seawead et al., 2025), several diffusion-based VR methods (Zhou et al., 2024; Yang et al., 2024; He et al., 2024a; Xie et al., 2025; Li et al., 2025a) have emerged, demonstrating remarkable performance. While fine-tuning on a diffusion prior (Rombach et al., 2022; Zhang et al., 2023) improves efficiency, these methods still inherit the inherent limitations of the diffusion prior, *i.e.*, inefficient autoencoder and inflexible resolution scalability as discussed by Wang et al. (2025). The most recent work (Wang et al., 2025) proposes to fully train a large diffusion transformer model with a shifted window attention and a casual video autoencoder, achieving impressive performance with relatively high efficiency. However, the need for tens of steps to sample a video still leads to unfriendly latency in real-world applications. By introducing APT (Lin et al., 2025) into diffusion-based VR, our approach is capable of achieving one-step VR with high quality, which, to the best of our knowledge, is among the earliest explorations of one-step diffusion-based VR.

**Diffusion Acceleration.** As discussed by Lin et al. (2025), most of the existing approaches either distill the deterministic probability flow learned by a diffusion teacher model using fewer steps (*i.e.*, deterministic methods) or approximate the same distribution of a diffusion teacher model (*i.e.*, distributional methods). Specifically, deterministic methods include progressive distillation (Salimans and Ho, 2022), consistency distillation (Luo et al., 2023a; Song et al., 2023; Song and Dhariwal, 2024; Lu and Song, 2025; Luo et al., 2023b), and rectified flow (Liu et al., 2023; 2024; Yan et al., 2024). Though these methods can be easily trained with simple regression loss, blurry results can be commonly observed with very few steps, *i.e.*, less than 8 steps (Luo et al., 2023a; Song et al., 2023; Luo et al., 2023b). In addition to directly predicting the outputs of the teacher model, distributional methods turn to adversarial training (Luo et al., 2025; Sauer et al., 2024b; Xu et al., 2024; Chen et al., 2025a; Kang et al., 2024b), score distillation (Yin et al., 2024a; Luo et al., 2023c), both (Sauer et al., 2024a; Yin et al., 2024b; Chadebec et al., 2025), and combining with deterministic methods (Kohler

et al., 2024; Lin et al., 2024b; Ren et al., 2024) to resemble the distribution of a teacher model. Most recent approaches (Lin et al., 2025; Xu et al., 2024) instead directly fine-tune a pre-trained diffusion model on real data with adversarial training, leading to superior performance with one-step sampling. While several acceleration approaches (Lin et al., 2025; Lin and Yang, 2024; Wang et al., 2024c; Zhai et al., 2024) have been extended to video generation, the one-step acceleration for video diffusion restoration is still underexplored, inspiring us to make an early attempt in this direction.

**One-step Restoration.** While conventional GAN-based real-world restoration approaches (Chan et al., 2022b; Zhang and Yao, 2024; Zhang et al., 2021; Wang et al., 2021; Zhou et al., 2022) can achieve one-step restoration, their poor generation ability usually leads to suboptimal results. To improve the sampling efficiency of diffusion-based approaches (Zhou et al., 2024; Yang et al., 2024; Wang et al., 2024a; Yu et al., 2024), ResShift (Yue et al., 2023; 2024) shifts the initial sampling distribution from a standard Gaussian distribution to the distribution of low-quality images, achieving a fast sampling of up to 4 steps. Recent advances further achieve one-step sampling via distillation (Wang et al., 2024b; Zhu et al., 2024; Li et al., 2025b; Dong et al., 2025; Xie et al., 2024; Sami et al., 2024; Noroozi et al., 2024; He et al., 2024b), adversarial training (Li et al., 2024), or tuning on a prior with additional trainable layers (Yue et al., 2025; Wu et al., 2024; Zhang et al., 2024). However, all these methods focus on image restoration and may not be suitable for VR due to the lack of temporal design and unsatisfactory generation quality. Compared with these methods, our method achieves one-step VR with substantially better quality, especially under high-resolution real-world scenarios.

# 3 METHODOLOGY

The objective of SeedVR2 is to perform one-step Video Restoration (VR) by upscaling an input video into a high-resolution output. SeedVR2 builds upon previous works (Wang et al., 2025; Lin et al., 2025), with preliminary concepts introduced in Sec.3.1.

The remainder of this section discusses VR-specific design improvements. Specifically, Sec.3.2 proposes an adaptive window attention mechanism to enhance test-time robustness for high-resolution videos. Sec. 3.3 explores one-step distillation within the adversarial post-training, and presents loss enhancements to improve training stability and model generalization.

## 3.1 PRELIMINARIES: DIFFUSION ADVERSARIAL POST-TRAINING

Diffusion Adversarial Post-Training (APT) (Lin et al., 2025) is a diffusion acceleration approach that converts a multi-step diffusion model to a one-step generator. There are mainly two training stages in APT, *i.e.*, deterministic distillation and Adversarial APT. During the deterministic distillation, a distilled model is first trained following discrete-time consistency distillation (Song et al., 2023; Song and Dhariwal, 2024) with mean squared error loss. The teacher model generates distillation supervision with a constant classifier-free guidance (Ho and Salimans, 2021) scale of 7.5 and a predefined negative prompt. As for adversarial training, the discriminator is first initialized by the pre-trained diffusion network, and then additional cross-attention-only transformer blocks are introduced to generate logits for loss calculation. To stabilize the adversarial training while avoiding higher-order gradient computation, APT proposes an approximated R1 loss (Roth et al., 2017) to regularize the discriminator, and the final loss for the discriminator is a non-saturating GAN loss (Goodfellow et al., 2014) combined with the approximated R1 loss. Our method employs a similar network architecture to APT, where both the generator and discriminator are diffusion transformers, as shown in Figure 2.

## 3.2 ADAPTIVE WINDOW ATTENTION

To improve the robustness of window attention for high-resolution inputs with arbitrary sizes, we propose an adaptive window attention mechanism that allows the window size to be dynamically adjusted to fit the input resolution, as shown in Figure 2. During training, given a video feature $X \in \mathbb{R}^{d_t \times d_h \times d_w \times d_c}$, where $d_h \times d_w = 45 \times 80$ (*i.e.*, the feature resolution under 720p), the window size of our attention is calculated accordingly as follows:

$$p_t = \left\lceil \frac{\min(d_t, 30)}{n_t} \right\rceil, \quad p_h = \left\lceil \frac{d_h}{n_h} \right\rceil, \quad p_w = \left\lceil \frac{d_w}{n_w} \right\rceil, \tag{1}$$

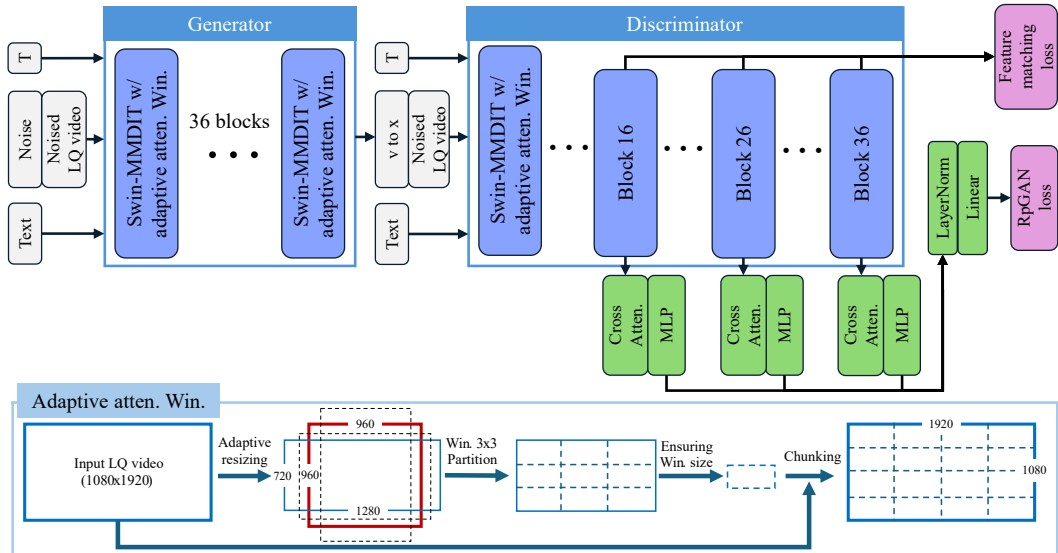

Figure 2: Model architecture and the adaptive attention window. We improve the Swin-MMDIT (Wang et al., 2025) with an adaptive window partition, *i.e.*, the window size is ensured via a $3 \times 3$ partition on the resized LQ input ($\text{Height} \times \text{Width} = 960 \times 960$). The features for calculating the feature matching loss are extracted before the cross-attention layers used in APT (Lin et al., 2025).

where $n_t$, $n_h$ and $n_w$ decide the number of windows along dimension $d_t$, $d_h$ and $d_w$, respectively. The ceiling function is represented as $\lceil \cdot \rceil$, and the term $\min(d_t, 30)$ sets an upper bound to $d_t$ to avoid the gap of sequence length between training and inference. Note that although the resolutions of our training data are around 720p, the aspect ratio of width and height can vary a lot, leading to various window sizes during training. Such a design ensures a better generalization ability toward inputs of different resolutions with diverse window sizes.

To further improve test-time robustness on high-resolution inputs, we introduce a resolution-consistent windowing strategy. Given a test-time video feature $\hat{X} \in \mathbb{R}^{\hat{d}_t \times \hat{d}_h \times \hat{d}_w \times \hat{d}_c}$, we first derive a spatial proxy resolution $\tilde{d}_h \times \tilde{d}_w$ that is consistent with the training resolution while maintaining the aspect ratio of the test input as follows:

$$\tilde{d}_h = \sqrt{d_h \times d_w \times \frac{\hat{d}_h}{\hat{d}_w}}, \quad \tilde{d}_w = \sqrt{d_h \times d_w \times \frac{\hat{d}_w}{\hat{d}_h}}, \tag{2}$$

where $d_h \times d_w = 45 \times 80$ is the training resolution. This ensures $\frac{\tilde{d}_h}{\tilde{d}_w} = \frac{\hat{d}_h}{\hat{d}_w}$ and $\tilde{d}_h \times \tilde{d}_w = d_h \times d_w$. The final window size for test-time attention is then obtained by substituting $(d_t, d_h, d_w)$ in Eq. (1) with $(\hat{d}_t, \tilde{d}_h, \tilde{d}_w)$. This adaptive partition strategy enhances consistency between training and testing configurations and substantially alleviates boundary artifacts in high-resolution predictions, as illustrated in Figure 4.

### 3.3 TRAINING PROCEDURES

Large-scale adversarial training is challenging. Benefiting from the low-quality condition in VR, we do not observe mode collapse (Goodfellow et al., 2014) when starting from adversarial training. However, undesired artifacts can be observed after training for thousands of iterations, indicating that the unstable training issue still exists. Our approach improves the training stability from the following two aspects, *i.e.*, distillation and loss.

**Progressive Distillation.** Directly adopting adversarial training to obtain a one-step model from an initial multi-step one may undermine the restoration ability of the model due to the large gap between the initial model and the target model. We conduct progressive distillation (Salimans and Ho, 2022) to alleviate such a problem. To be specific, we start with the teacher model initialized from SeedVR (Wang et al., 2025) with 64 sampling steps and progressively distill the student model to one step with a distillation stride of 2. Each distillation procedure takes about 10K iterations with a

simple mean squared error loss. We also progressively increase the temporal length of the training data from images to video clips with a diverse number of frames during adversarial training, leading to robust VR performance toward videos with various lengths, including images. Benefiting from such a training strategy, we further obtain a 3B model distilled from the original 7B one, achieving comparable performance with only half of the model size.

**Loss Improvement.** Inspired by R3GAN (Huang et al., 2024), we first replace the non-saturating GAN loss (Goodfellow et al., 2014) used in APT by a RpGAN loss (Jolicoeur-Martineau, 2019) to avoid the potential mode dropping problem. We further introduce an approximate R2 regularization to penalize the gradient norm of $D$ on fake data while supporting modern deep learning software stacks:

$$\mathcal{L}_{aR2} = \|D(\hat{\boldsymbol{x}}, c) - D(\mathcal{N}(\hat{\boldsymbol{x}}, \sigma\mathbf{I}), c)\|_2^2, \tag{3}$$

where $\hat{\boldsymbol{x}}$ denotes the sample prediction converted from the velocity field output from the model, $c$ is the text condition, $\sigma$ controls the variance of the perturbing Gaussian noise, and $\mathbf{I}$ represents the identity matrix. We observe that the above loss improvements ensure a more stable training without mode collapse after training for thousands of iterations.

Besides GAN loss, L1 loss and LPIPS loss are commonly used in VR for perception-distortion tradeoff (Blau and Michaeli, 2018). However, to compute LPIPS loss, we have to first decode the prediction from the latent space to pixel space, leading to an unaffordable computational cost in our scenario. Instead of LPIPS loss, we propose to adopt a feature matching loss via directly extracting features from the discriminator for efficient loss calculation. Specifically, we extract the features of predictions and ground-truths before the attention-only transformer blocks (*i.e.*, the 16th, 26th, and 36th blocks of the transformer backbone) of the discriminator. Then, our feature matching loss $\mathcal{L}_F$ can be written as:

$$\mathcal{L}_F = \frac{1}{3} \sum_{i=16,26,36} \|D_i^F(\hat{\boldsymbol{x}}, c) - D_i^F(\boldsymbol{x}, c)\|_1, \tag{4}$$

where $D_i^F(\cdot)$ denotes the feature from the i-th block of the discriminator. By default, we set the loss weight as $1.0$ for L1 loss, feature matching loss, and GAN loss when updating the generator. When updating the discriminator, we apply a weight of $1.0$ for GAN loss and the weights of the approximate R1 and R2 regularization are both $1000$. Note that the discriminator is fixed when updating the generator. In this way, the discriminator in our feature matching loss acts in a similar way to the VGG network (Simonyan and Zisserman, 2015) in LPIPS loss. Besides, the feature matching loss should also work with other GAN losses (Goodfellow et al., 2014; Arjovsky et al., 2017; Mao et al., 2017; Gulrajani et al., 2017) to further stabilize adversarial training for restoration tasks.

## 4 EXPERIMENTS

**Implementation Details.** We train SeedVR2 on 72 NVIDIA H100-80G GPUs with around 100 frames of 720p per batch with sequence parallel (Korthikanti et al., 2023) and data parallel (Li et al., 2020). Each stage of training takes about one day. We first train a 7B SeedVR model (Wang et al., 2025) from scratch following the new attention design in this paper. Then, we initialize the model parameters from 7B SeedVR model and follow the training strategies discussed in Sec. 3.3 for our SeedVR2 models. We mostly follow the training settings in APT (Lin et al., 2025) for adversarial training. We follow UAV (Zhou et al., 2024) to synthesize about 10M image pairs and 5M video pairs for training. During the distillation, loss is calculated on the vector field following Flow matching (Lipman et al., 2023). Both teacher and student models adopt the linear noise schedule with a timestep between 0 and 999. The teacher model uses the Euler sampler during training with a CFG scale of 7.5 for 64 timesteps and 1.0 for others. We adopt AdamW (Kingma and Ba, 2014) with a weight decay of 0.01 as optimizer, and the learning rate is set to $1 \times 10^{-6}$.

**Experimental Settings.** Following previous work (Zhou et al., 2024), we evaluate synthetic benchmarks, including SPMCS (Yi et al., 2019), UDM10 (Tao et al., 2017), REDS30 (Nah et al., 2019), and YouHQ40 (Zhou et al., 2024), applying the same degradation settings as in training. The test resolution is 720p with an upscaling factor of $4$. Furthermore, we assess performance on the commonly used real-world dataset (VideoLQ (Chan et al., 2022b)) and a self-collected AIGC dataset (AIGC28), which comprises 28 AI-generated videos with diverse resolutions and scenes. We employ a range of metrics to assess both frame-level and overall video quality. For synthetic pair datasets, we adopt

Table 1: Quantitative comparisons on VSR benchmarks from diverse sources, *i.e.*, synthetic (SPMCS, UDM10, REDS30, YouHQ40), real (VideoLQ), and AIGC (AIGC28) data. The best and second performances are marked in red and orange, respectively.

| Datasets | Metrics | RealViformer | MGLD-VSR | UAV | VEnhancer | STAR | SeedVR-7B | Ours-3B | Ours-7B |
|---|---|---|---|---|---|---|---|---|---|
| SPMCS | PSNR ↑ | 24.19 | 23.41 | 21.69 | 18.20 | 22.58 | 20.78 | 22.97 | 22.90 |
| | SSIM ↑ | 0.663 | 0.633 | 0.519 | 0.507 | 0.609 | 0.575 | 0.646 | 0.638 |
| | LPIPS ↓ | 0.378 | 0.369 | 0.508 | 0.455 | 0.420 | 0.395 | 0.306 | 0.322 |
| | DISTS ↓ | 0.186 | 0.166 | 0.229 | 0.194 | 0.229 | 0.166 | 0.131 | 0.134 |
| UDM10 | PSNR ↑ | 26.70 | 26.11 | 24.62 | 21.48 | 24.66 | 24.29 | 25.61 | 26.26 |
| | SSIM ↑ | 0.796 | 0.772 | 0.712 | 0.691 | 0.747 | 0.731 | 0.784 | 0.798 |
| | LPIPS ↓ | 0.285 | 0.273 | 0.323 | 0.349 | 0.359 | 0.264 | 0.218 | 0.203 |
| | DISTS ↓ | 0.166 | 0.144 | 0.178 | 0.175 | 0.195 | 0.124 | 0.106 | 0.101 |
| REDS30 | PSNR ↑ | 23.34 | 22.74 | 21.44 | 19.83 | 22.04 | 21.74 | 21.90 | 22.27 |
| | SSIM ↑ | 0.615 | 0.578 | 0.514 | 0.545 | 0.593 | 0.596 | 0.598 | 0.606 |
| | LPIPS ↓ | 0.328 | 0.271 | 0.397 | 0.508 | 0.487 | 0.340 | 0.350 | 0.337 |
| | DISTS ↓ | 0.154 | 0.097 | 0.181 | 0.229 | 0.229 | 0.122 | 0.135 | 0.127 |
| YouHQ40 | PSNR ↑ | 23.26 | 22.62 | 21.32 | 18.68 | 22.15 | 20.60 | 22.10 | 22.46 |
| | SSIM ↑ | 0.606 | 0.576 | 0.503 | 0.509 | 0.575 | 0.546 | 0.595 | 0.600 |
| | LPIPS ↓ | 0.362 | 0.356 | 0.404 | 0.449 | 0.451 | 0.323 | 0.284 | 0.274 |
| | DISTS ↓ | 0.193 | 0.166 | 0.196 | 0.175 | 0.213 | 0.134 | 0.122 | 0.110 |
| VideoLQ | NIQE ↓ | 4.153 | 3.864 | 4.079 | 5.122 | 5.915 | 4.933 | 4.687 | 4.948 |
| | MUSIQ ↑ | 54.65 | 53.49 | 52.90 | 42.66 | 40.50 | 48.35 | 51.09 | 45.76 |
| | CLIP-IQA ↑ | 0.411 | 0.333 | 0.386 | 0.269 | 0.243 | 0.258 | 0.295 | 0.257 |
| | DOVER ↑ | 7.035 | 8.109 | 6.975 | 7.985 | 6.891 | 7.416 | 8.176 | 7.236 |
| AIGC28 | NIQE ↓ | 3.994 | 4.049 | 4.541 | 4.176 | 5.004 | 4.294 | 3.801 | 4.015 |
| | MUSIQ ↑ | 62.82 | 60.98 | 62.79 | 60.99 | 55.59 | 56.90 | 62.99 | 59.97 |
| | CLIP-IQA ↑ | 0.647 | 0.570 | 0.653 | 0.461 | 0.435 | 0.453 | 0.561 | 0.497 |
| | DOVER ↑ | 11.66 | 14.27 | 13.09 | 15.31 | 14.82 | 14.77 | 15.77 | 15.55 |

full-reference metrics, including PSNR, SSIM, LPIPS (Zhang et al., 2018b), and DISTS (Ding et al., 2020). For real-world and AI-generated content (AIGC) test data, where ground truth is unavailable, we rely exclusively on no-reference metrics, *i.e.*, NIQE (Mittal et al., 2012), CLIP-IQA (Wang et al., 2023), MUSIQ (Ke et al., 2021), and DOVER (Wu et al., 2023)[1]. To ensure test efficiency, the maximum output resolution is constrained to 1080p, with duration unchanged.

## 4.1 COMPARISON WITH EXISTING METHODS

**Quantitative Comparisons.** We compare our approach with all state-of-the-art real-world video restoration approaches. For diffusion-based methods, *i.e.*, MGLD-VSR (Yang et al., 2024), UAV (Zhou et al., 2024), VEnhancer (He et al., 2024a), STAR (Xie et al., 2025), SeedVR-7B (Wang et al., 2025), we adopt 50 sampling steps with a wavelet color fix post-processing (Wang et al., 2024a), and keep other official settings unchanged. As shown in Table 1, our approach demonstrates superior performance in terms of perceptual metrics such as LPIPS and DISTS on synthetic benchmarks including SPMCS, UDM10 and YouHQ40. Note that RealViformer (Zhang and Yao, 2024) and MGLD-VSR involve REDS in the train data, leading to high performance on the corresponding test set. As for real-world benchmarks, our method achieves comparable performance compared with other diffusion-based methods on VideoLQ and further obtains the highest NIQE, MUSIQ and DOVER scores on AIGC28, demonstrating our effectiveness.

**Qualitative Comparisons.** As observed in several previous studies (Yu et al., 2024; Blau and Michaeli, 2018; Yue and Loy, 2024; Gu et al., 2022), existing image and video quality assessment metrics do not perfectly align with human perception. For example, non-reference metrics such as MUSIQ and CLIP-IQA prefer sharp results but may ignore the quality of details. We notice that such a phenomenon becomes more evident under high resolutions, *e.g.*, 1080p. As shown in Figure 3, while our method does not show dominant metric performance on VideoLQ, the results generated by our approach are comparable to SeedVR and outperform other baselines by a large margin.

**User Study.** To further validation, we follow APT (Lin et al., 2025) to conduct a GSB test, *i.e.*, the preference score is calculated as $\frac{G-B}{G+S+B}$, where G is the number of good samples preferred by the subjects, B is the bad samples not preferred, and S denotes the number of similar samples without preference. Thus, the score ranges from $-100\%$ to $100\%$ and $0\%$ indicates equal performance.

---

[1]We adopt the technical score ranging from 0 to 100 following the official code.

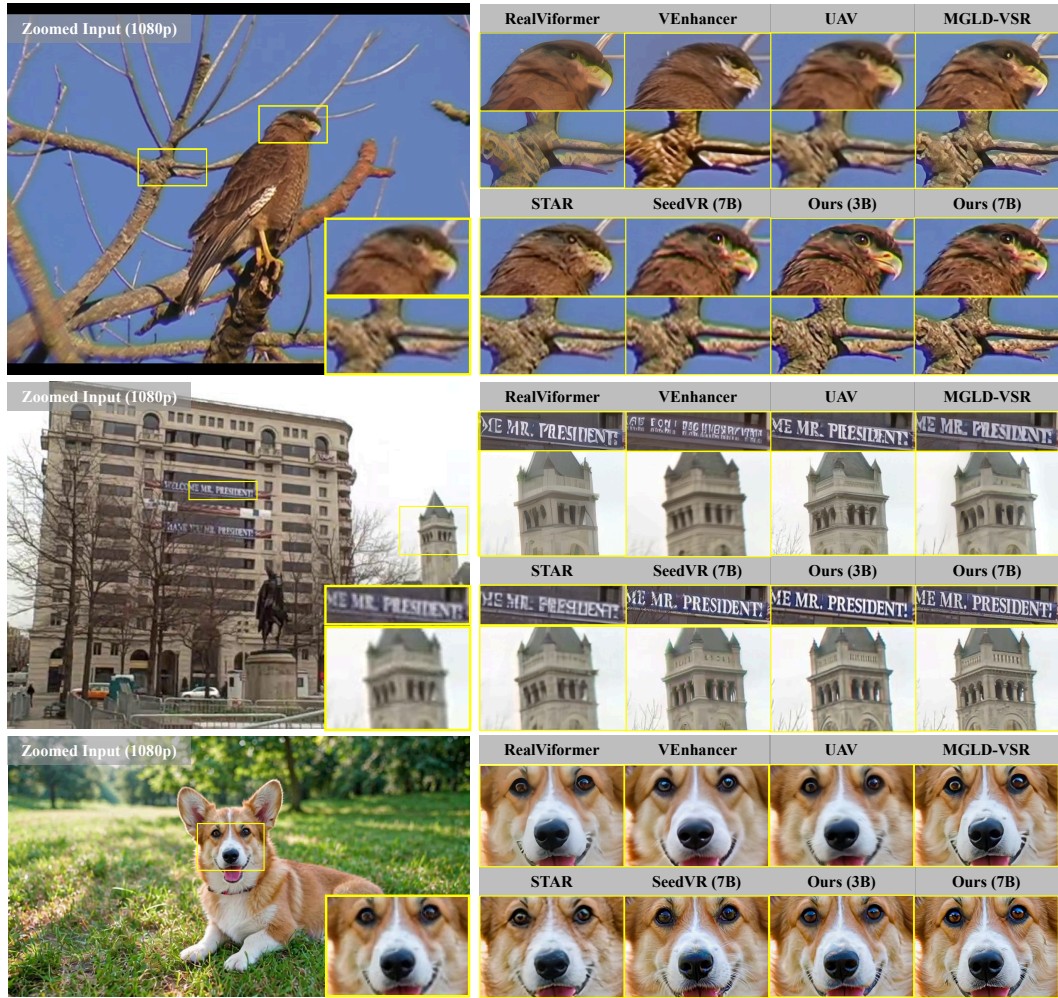

Figure 3: Qualitative comparisons on both real-world (Chan et al., 2022b) and AIGC videos. With a single sampling step, our method achieves comparable performance to SeedVR, further excelling other baselines with superior restoration capabilities, *i.e.*, successfully removing the degradations while maintaining the textures of the bird, text, building, and the dog's face (**Zoom-in for best view**).

We randomly select 25 samples from VideoLQ (Chan et al., 2022b) and AIGC28, respectively, resulting in 50 LQ videos for test in total. We set our approach (7B) as the datum and compare it with existing methods (Wang et al., 2025; Zhou et al., 2024; Yang et al., 2024; He et al., 2024a; Xie et al., 2025; Zhang and Yao, 2024). Given the LQ videos as reference, three experts are asked to evaluate the generated video quality from the following three criteria: *visual fidelity*, *visual quality* and *overall quality*. The visual fidelity measures the content similarity between the LQ reference and the output. The visual

Table 2: Our one-step video restoration compared to existing methods.

| Methods-{Steps} | Visual Fidelity | Visual Quality | Overall Quality |
|---|---|---|---|
| RealViformer-1 | +2% | -38% | -32% |
| VEnhancer-50 | -82% | -86% | -94% |
| UAV-50 | 0% | -26% | -26% |
| MGLD-VSR-50 | 0% | -12% | -12% |
| STAR-50 | +4% | -22% | -24% |
| SeedVR-7B-50 | +2% | +10% | +10% |
| Ours-3B-1 | 0% | +16% | +16% |
| Ours-7B-1 | 0% | 0% | 0% |

quality focuses on the realism of the generated results. The overall quality indicates the final preference after taking the above two factors as well as temporal consistency into account. The subjects are given a pair of videos generated by different methods each time and asked to make their preferences for each criterion.

As shown in Table 2, our approach is comparable to the multi-step SeedVR and clearly excels other approaches with better visual quality, aligning with the visual results shown in Figure 3. Particularly, VEnhancer focuses on generative restoration, thus showing poor fidelity in real-world VR scenarios.

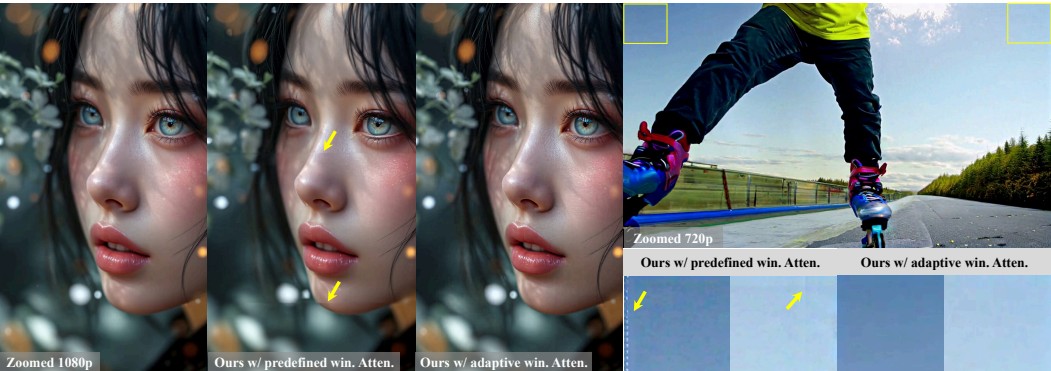

Figure 4: Comparisons of the window attention with a predefined size (*i.e.*, ours w/ predefined win. atten.) and our adaptive window attention (*i.e.*, ours w/ adaptive win. atten.). Boundary artifacts can be observed on high-resolution restoration with the predefined-size window attention.

Restricted by the limited generative capability of the diffusion prior, existing approaches (Zhou et al., 2024; Yang et al., 2024; Xie et al., 2025) tend to generate inferior results with high-resolution inputs, indicating the necessity to train a large VR model without relying on the fixed prior. While our methods, *i.e.*, ours-3B and ours-7B, clearly outperform several baselines (Zhou et al., 2024; Yang et al., 2024; He et al., 2024a; Xie et al., 2025; Zhang and Yao, 2024), the performance between these two models is different. Specifically, ours-3B receives more preference from the subjects than ours-7B, aligning with the results in Table 1. Recall that ours-3B is distilled from the 7B initial model. Such a performance gain may indicate the effectiveness of the distillation stage. And we believe our 7B model could receive further improvement with the scaling of computational resources.

## 4.2 ABLATION STUDY

**The Effect of Adaptive Window Attention.** We first examine the effectiveness of the proposed adaptive window attention. We train the model with the predefined-size window attention and the proposed adaptive window attention, respectively. Both models share the same training settings for 20k iterations. As shown in Figure 4, when generating high-resolution (*e.g.*, 1080p) results, window boundary inconsistency can be observed with a predefined-size attention window. We conjecture that such drawbacks indicate the limited model capability of handling overlapping windows, which is associated with the improper setting of the window size compared to the training resolutions. Specifically, applying a $64 \times 64$ window over the compressed latent with a downsampling factor of 8 makes the model insufficiently trained on window-overlapping cases, which are rare on the 720p training pairs. Moreover, we find that the diffusion transformer with RoPE embeddings (Su et al., 2024) shows more robust performance across a range of resolutions after training on data with various sizes. Shifting to the window attention with mostly predefined window size (Wang et al., 2025) may weaken the generalization ability on other window sizes, *i.e.*, the variable-sized windows near the boundary as shown in Figure 4. We show that the proposed adaptive window attention significantly improves the model robustness by mitigating the aforementioned failure cases.

Table 3: Ablation study on training losses and progressive training. All baselines are trained on 72 NVIDIA H100-80G cards for 20k iterations. The comparison is conducted on YouHQ40 (Zhou et al., 2024). Note that the first four baselines are trained w/o progressive training, while the last one is trained following our proposed method, but with different iterations for fair comparison.

| Metrics | Non-satu. + R1 | RpGAN + R1 + R2 | RpGAN + R1 + R2 + L1 | RpGAN + R1 + R2 + L1 + LF | w/ Prog. Training |
|---|---|---|---|---|---|
| PSNR ↑ | 22.55 | 22.56 | 22.91 | 22.91 | 23.96 |
| SSIM ↑ | 0.612 | 0.603 | 0.616 | 0.620 | 0.667 |
| LPIPS ↓ | 0.310 | 0.278 | 0.251 | 0.244 | 0.227 |
| DISTS ↓ | 0.136 | 0.109 | 0.099 | 0.092 | 0.097 |

**The Effect of Losses and Progressive Distillation.** Training a large-scale GAN can be challenging due to its unstable nature. We verify the significance of various losses used in our method. We train each baseline with different loss combinations for 20k iterations and keep other settings the same. As

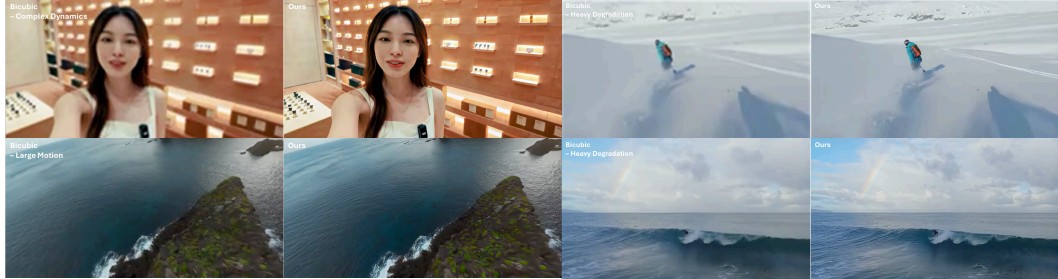

Figure 5: Qualitative results on real-world videos with challenging conditions, *e.g.*, heavy degradations, large motion, and complex dynamics. Video comparison can be found in the demo video in the supplementary materials **(Zoom-in for best view)**.

shown in Table 3, compared with the vanilla loss used in APT (Lin et al., 2025) (*i.e.*, non-saturating GAN loss (Goodfellow et al., 2014) + R1), the model trained with RpGAN (Jolicoeur-Martineau, 2019), R1 and R2 losses demonstrate significant improvement on perceptual metrics such as LPIPS and DISTS. We further observe a more stable training procedure without mode collapse, which exists under the settings of APT after long training. Besides, the adoption of L1 loss and the proposed feature matching loss both improve the metric performance, indicating the significance of these losses for restoration tasks. In practice, we notice that a large loss weight of L1 loss and feature matching loss improves the fidelity, but may lead to mildly over-smooth results compared with assigning a large weight to the GAN loss. Such an observation is consistent with the perception-distortion theory (Blau and Michaeli, 2018). As a result, we reduce the loss weight of L1 loss and the feature matching loss to 0.1 for the final model to enable better visual quality as reported in Sec. 4.1. Finally, as indicated in Table 3, applying a progressive distillation is necessary to maintain a strong restoration ability, which is expected since the distillation effectively minimizes the gap between the initial model and the one-step adversarial training.

### 4.3 LIMITATIONS

The effectiveness of our SeedVR2 can be verified under challenging conditions as shown in Figure 5. However, we further identify several limitations of current SeedVR2 in practice. While our one-step method significantly saves time during sampling, the causal video VAE requires over 4x more time to encode and decode a video compared to the naive VAE commonly used by existing methods (Zhou et al., 2024; Yang et al., 2024; He et al., 2024a; Xie et al., 2025). In addition, when dealing with a 720p video with 100 frames, the casual video VAE takes over 95% of the total time. Enhancing the efficiency of the video VAE while maintaining performance is a worthwhile direction for future work.

Besides the VAE efficiency, we notice that our method is sometimes not robust to heavy degradations and very large motions, and shares some failure cases with existing methods, *e.g.*, it may fail to fully remove degradations or generate visually unpleasing details. Moreover, due to the strong generation ability, SeedVR2 tends to overly generate details on inputs with very light degradations, *e.g.*, 720p AIGC videos, leading to oversharpened results occasionally. Thus, we have to tune the model with careful hyperparameter settings. Improving the robustness towards complex real-world degradations and ensuring a satisfactory lower bound of performance remains a challenge for future work.

## 5 CONCLUSION

In this paper, we have presented SeedVR2, an early exploration on the one-step diffusion transformer model toward real-world restoration. SeedVR2, building on the adversarial post-training with a pre-trained diffusion model as initialization, tackles one-step video restoration through tailored designs such as an adaptive window attention and several training enhancements, along with a feature matching loss, which are crucial for stabilizing large-scale adversarial training and improving the restoration performance. Despite the large parameter size, SeedVR2 is over four times faster than existing multi-step diffusion VR methods, with comparable or even superior performance as shown by our experiments. In the future, we will improve the robustness of SeedVR2 towards complex degradations and further optimize the parameter size to facilitate real-time applications. We believe our proposed SeedVR2 could provide useful insights for future works.

## 6 ETHICS STATEMENT

Our approach is likely to push forward the progress of restoration applications toward real-world image and video restoration. Specifically, our approach may inspire future work to develop fast restoration methods with strong performance. The release of our model weights and code could further contribute to the restoration community in developing their own large restoration models. Of particular concern is the misconduct of applying our method to enhance illegal content, such as NSFW. To mitigate this risk, we plan to include the corresponding detection tool in our public code to restrict the use of our method.

## 7 REPRODUCIBILITY STATEMENT

Our implementation is based on PyTorch 2.4.0 with CUDA 12.4. All the referenced software and models used in this paper are publicly available. Our code and model checkpoints will be released for reproducibility upon acceptance.

## ACKNOWLEDGMENTS

This research is supported by the National Research Foundation, Singapore under its AI Singapore Programme (AISG Award No: AISG2-PhD-2022-01-033[T]). We sincerely thank Zhibei Ma for data processing, Jiashi Li for hardware maintenance. We also thank Feng Cheng for performance feedback, Yi Li, Fangyuan Kong and Rui Zhang for training suggestions.

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

# A    APPENDIX

## A.1    USE OF LARGE LANGUAGE MODELS

The large language models (LLMs), i.e., GPT-4o and Gemini 2.5 Pro, are solely used for polishing some paragraphs in this paper for clarity of expression and avoidance of minor grammar errors. They are not involved in any aspects related to the research contributions of this paper.

## A.2    ADDITIONAL EVALUATIONS

Table 4: Additional full-reference metrics on synthetic benchmarks (SPMCS, UDM10, REDS30, YouHQ40). The best and second performances are marked in red and orange, respectively.

| Datasets | Metrics | Bicubic | RealViformer | MGLD-VSR | UAV | VEnhancer | STAR | SeedVR-7B | **Ours**-3B | **Ours**-7B |
|---|---|---|---|---|---|---|---|---|---|---|
| SPMCS | NIQE ↓ | 9.105 | 3.431 | 3.315 | 3.272 | 4.328 | 5.659 | 3.671 | 3.862 | 3.668 |
| | MUSIQ ↑ | 24.65 | 62.09 | 65.25 | 65.01 | 54.94 | 37.74 | 70.03 | 63.52 | 63.37 |
| | CLIP-IQA ↑ | 0.3448 | 0.4239 | 0.4948 | 0.5074 | 0.3341 | 0.2346 | 0.4690 | 0.4736 | 0.4367 |
| | DOVER ↑ | 0.9490 | 7.664 | 8.471 | 6.237 | 7.807 | 3.728 | 10.02 | 9.754 | 9.333 |
| | $E^*_{warp}$ ↑ | 0.395 | 0.655 | 1.414 | 2.188 | 0.768 | 0.479 | 1.797 | 0.715 | 0.845 |
| | VMAF ↑ | 5.24 | 20.43 | 34.47 | 27.19 | 16.96 | 14.13 | 43.05 | 39.96 | 39.86 |
| UDM10 | NIQE ↓ | 8.625 | 3.922 | 3.814 | 3.494 | 4.883 | 5.273 | 4.025 | 4.545 | 4.518 |
| | MUSIQ ↑ | 22.70 | 55.60 | 58.01 | 58.31 | 46.37 | 38.62 | 62.49 | 56.59 | 53.28 |
| | CLIP-IQA ↑ | 0.3377 | 0.3972 | 0.4430 | 0.4583 | 0.3035 | 0.2338 | 0.4404 | 0.3695 | 0.3459 |
| | DOVER ↑ | 1.592 | 7.259 | 7.717 | 9.238 | 8.087 | 5.374 | 10.63 | 9.652 | 8.907 |
| | $E^*_{warp}$ ↑ | 0.506 | 0.630 | 1.412 | 1.558 | 0.771 | 0.775 | 1.953 | 0.777 | 0.830 |
| | VMAF ↑ | 12.40 | 36.24 | 45.85 | 29.45 | 19.76 | 21.37 | 52.76 | 52.19 | 52.13 |
| REDS30 | NIQE ↓ | 9.058 | 3.032 | 2.550 | 2.561 | 4.615 | 5.548 | 3.462 | 4.034 | 3.825 |
| | MUSIQ ↑ | 19.43 | 58.60 | 62.28 | 56.39 | 37.95 | 30.08 | 57.80 | 50.90 | 50.79 |
| | CLIP-IQA ↑ | 0.2538 | 0.3920 | 0.4442 | 0.3978 | 0.2445 | 0.2109 | 0.3019 | 0.2601 | 0.2622 |
| | DOVER ↑ | 1.459 | 5.229 | 6.544 | 5.234 | 5.549 | 3.843 | 6.795 | 6.365 | 6.002 |
| | $E^*_{warp}$ ↑ | 0.852 | 1.619 | 3.897 | 5.366 | 2.131 | 1.841 | 4.854 | 4.010 | 4.084 |
| | VMAF ↑ | 9.21 | 28.32 | 38.05 | 28.16 | 12.05 | 18.88 | 43.28 | 41.73 | 39.93 |
| YouHQ40 | NIQE ↓ | 9.241 | 3.172 | 3.255 | 3.000 | 4.161 | 5.470 | 3.288 | 3.722 | 3.378 |
| | MUSIQ ↑ | 24.78 | 61.88 | 63.95 | 64.45 | 54.18 | 37.40 | 69.68 | 63.34 | 66.01 |
| | CLIP-IQA ↑ | 0.3162 | 0.4376 | 0.5085 | 0.4710 | 0.3518 | 0.2466 | 0.5132 | 0.4649 | 0.4687 |
| | DOVER ↑ | 1.276 | 9.483 | 10.50 | 9.957 | 11.44 | 3.817 | 13.42 | 13.34 | 12.98 |
| | $E^*_{warp}$ ↑ | 0.606 | 0.996 | 2.057 | 3.249 | 1.284 | 0.785 | 3.210 | 1.376 | 1.725 |
| | VMAF ↑ | 4.82 | 20.34 | 30.74 | 16.29 | 12.25 | 13.02 | 34.56 | 36.49 | 34.07 |

**Additional Metrics.** Due to the limited space, we provide additional metrics, i.e., NIQE (Mittal et al., 2012), MUSIQ (Ke et al., 2021), CLIP-IQA (Wang et al., 2023), DOVER (Wu et al., 2023), warping error (Zhou et al., 2024), and VMAF (Li et al., 2016) on the synthetic datasets (Zhou et al., 2024; Yi et al., 2019; Tao et al., 2017; Nah et al., 2019) for further evaluation across various baselines. As shown in Table 4, our proposed one-step approach consistently achieves comparable performance with other multi-step methods (50 steps) in terms of CLIP-IQA, MUSIQ on three of the four datasets, and DOVER, VMAF across all the four datasets. Note that the good performance of MGLD-VSR on

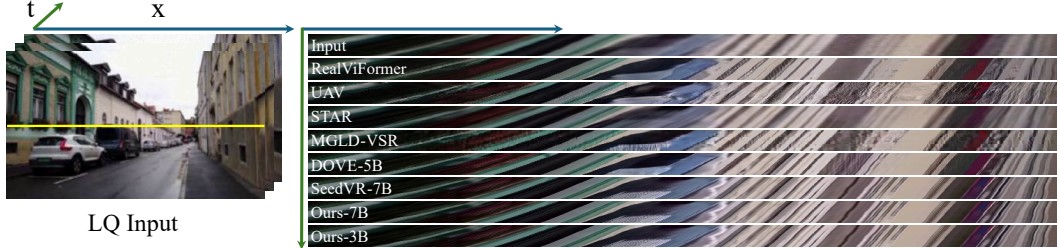

Figure 6: We follow UAV (Zhou et al., 2024) to examine a row and track changes over time. Our method can achieve satisfactory temporal consistency with clear edges and patterns along the temporal dimension. **(Zoom-in for best view)**.

Table 5: Quantitative comparisons on real data between progressive distillation from a teacher model and ours.

| Datasets | Methods | NIQE ↓ | MUSIQ ↑ | CLIP-IQA ↑ | DOVER ↑ |
|---|---|---|---|---|---|
| VideoLQ | Prog. Distill. | 5.365 | 45.57 | 0.230 | 6.609 |
| | Ours-3B | 4.687 | 51.09 | 0.295 | 8.176 |
| AIGC28 | Prog. Distill. | 4.857 | 58.85 | 0.416 | 13.11 |
| | Ours-3B | 3.801 | 62.99 | 0.561 | 15.77 |

REDS30 is expected since this method is trained on the training set of REDS (Nah et al., 2019) while other methods do not include such data for training. While our method does not show advantages in warping error, it is noticeable that the naive bicubic upsampling shows the best performance over all other existing baselines (i.e., 0.395, 0.506, 0.852, 0.606 on SPMCS, UDM10, REDS30, and YouHQ40, respectively). The reason is that the warping error measures the accuracy of the optical flow between the predicted frames and the ground-truth ones. Given the ill-posed nature of the restoration problem, the predictions may not perfectly align with the assigned ground-truth in synthetic datasets. Such a phenomenon can be more pronounced for models with stronger generative ability like ours. Thus, we argue that such a metric does not accurately reflect the temporal quality of our approach. Moreover, our method shows superior performance on other video metrics such as VMAF and DOVER, which both take temporal stability and temporal coherence into consideration. The user study in Table 2 and the demos provided in this supplementary material further demonstrate the high perceptual quality of our generated results.

**Temporal Profile.** We follow UAV (Zhou et al., 2024) to provide the visualization of the temporal profile in Figure 6. Compared with other baselines, our method can achieve satisfactory temporal consistency with clear edges and patterns along the temporal dimension.

**Additional Visual Results.** We further show additional comparisons in Figure 7. For more image and video demos generated by our SeedVR2, please refer to our project page: https://iceclear.github.io/projects/seedvr2/ for details.

**Adversarial Training vs. Distillation.** While distillation (Salimans and Ho, 2022) is a widely used strategy for diffusion acceleration, the effectiveness of distillation from a pre-trained teacher model in video restoration is still in doubt without further evidence. Without a teacher model, we observed that the model trained with our proposed adversarial approach presents the potential to surpass the initial model, which theoretically cannot be achieved with naive distillation from a teacher model.

Besides the inherent performance constraint imposed by the teacher model, the loss functions commonly used in distillation further limit the performance gains of the student model. In contrast, our method is based on adversarial training, which have been widely recognized as more effective for image restoration compared to non-adversarial methods, since SRGAN (Ledig et al., 2017). Compared to adversarial approaches, distillation from a teacher model resembles non-adversarial methods that rely on L1 or L2 losses, which tend to produce over-smoothed results with a performance upper bound from the teacher model, especially for sampling with very few steps. To further strengthen such a claim, we present a comparison between the distillation baseline without adversarial post-training and our proposed approach. Both of the baselines are trained for the same iterations for fair comparison. Both quantitative results in Table 5 and qualitative visualizations in Figure 8 on real datasets demonstrate the advantages of avoiding the constraint of a teacher model.

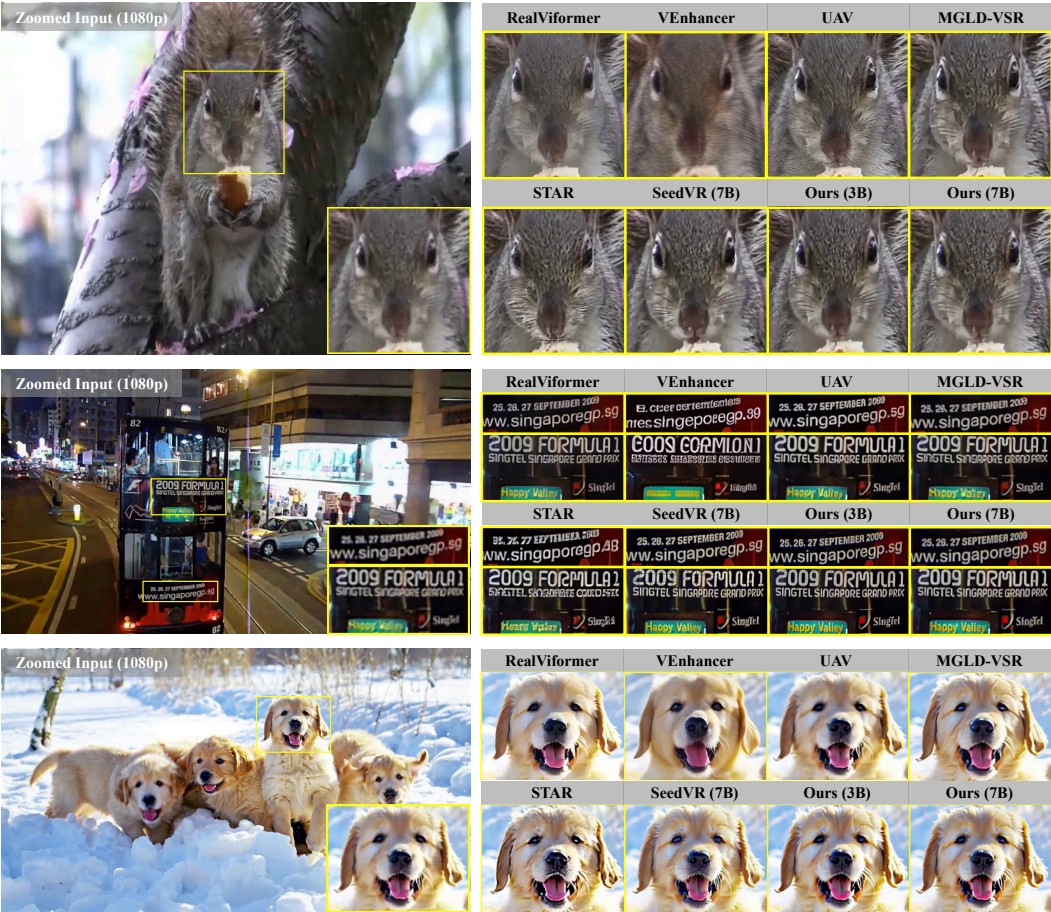

Figure 7: Qualitative comparisons on both real-world (Chan et al., 2022b) and AIGC videos. It is noticeable that the GAN-based approach (Zhang and Yao, 2024) generates blurry results due to limited generation ability. Previous multi-step diffusion-based VR (Zhou et al., 2024; Yang et al., 2024; He et al., 2024a; Xie et al., 2025) either fail to restore the low-quality video with faithful details or tend to generate oversharpened details. Even with a single sampling step, our approach clearly excels over these methods with a large margin. (**Zoom-in for best view**).

Table 6: Comparison of model parameters and inference time on 720p video with 100 frames.

| Metrics | VEnhancer | UAV | MGLD-VSR | STAR | SeedVR-7B | Ours-3B | Ours-7B |
|---|---|---|---|---|---|---|---|
| Number of Parameters (M) (Generator only) | 2044.8 | 691.0 | 1430.8 | 2041.0 | 8239.6 | 3391.5 | 8239.6 |
| Inference time s/video ($100 \times 768 \times 1344$) | 2029.2 | 1284.5 | 1181.0 | 2326.0 | 1284.8 | 269.0 | 299.4 |

## A.3 PARAMETER SIZE AND INFERENCE SPEED

We provide a detailed statistic regarding the number of parameters and inference time in Figure 1 of the main paper. We apply 50 sampling steps and keep other settings the same as the official repository for other baselines to maintain high-quality generation results of these methods. The results are listed in Table 6 for reference.

## A.4 COMPARISON WITH CONCURRENT WORK

Before the submission, we further notice that there is a concurrent work (Chen et al., 2025b) targeting at one-step video restoration. Specifically, the proposed method, named as DOVE (Chen et al., 2025b) proposes a two-stage strategy where the VAE model is first adapted to achieve the restoration mapping

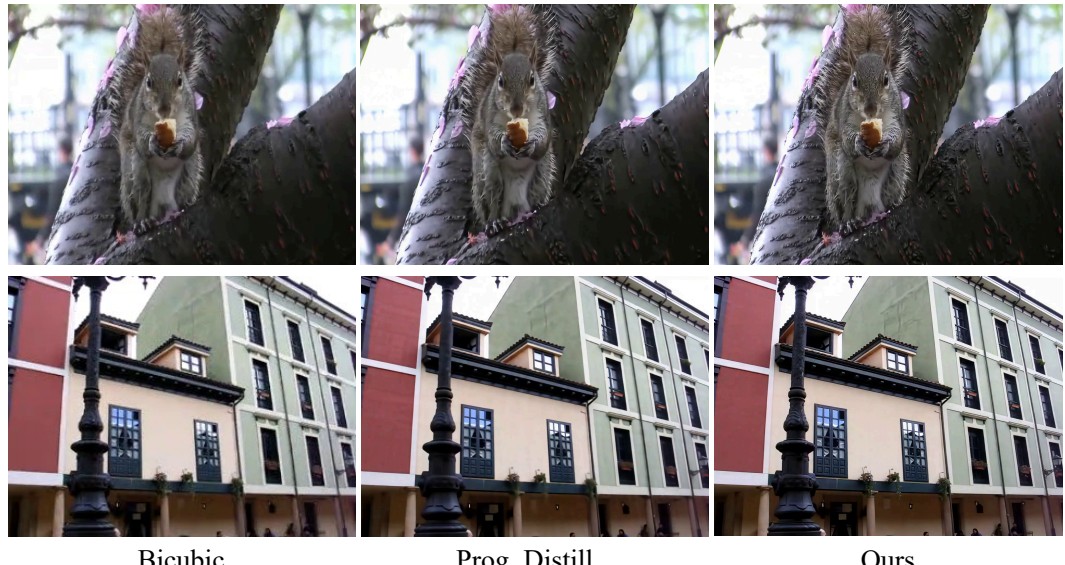

|  Bicubic | Prog. Distill. | Ours |

Figure 8: We present the visual comparison between progressive distillation (Salimans and Ho, 2022) and our proposed method based on adversarial training. The sharper edges and finer details indicate the potential of our proposed approach to avoid the constraint of a teacher model compared with the naive distillation. **(Zoom-in for best view)**.

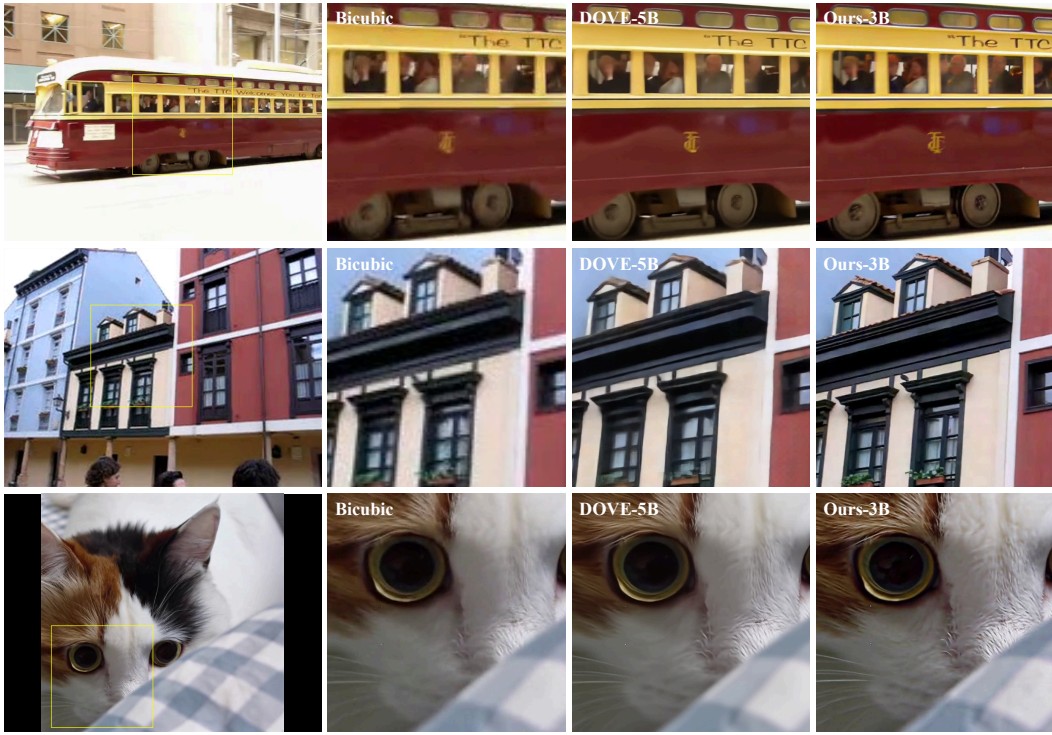

Figure 9: Comparison with DOVE-5B (Chen et al., 2025b) on real-world data. Our 3B model is capable of generating faithful details. **(Zoom-in for best view)**.

between low-quality input and the ground truth, and the diffusion transformer model initialized from CogVideoX1.5 (Yang et al., 2025) is refined in the pixel domain with a combination of L1 loss, DISTS loss, and a frame difference loss. Our proposed approach considers an orthogonal path to achieve one-step acceleration with adversarial training and avoid training in the pixel domain for high efficiency under large-scale training. We make a comparison with DOVE-5B on real-world

Table 7: Quantitative comparisons on real data between DOVE-5B and Ours-3B.

| Datasets | Methods | NIQE ↓ | MUSIQ ↑ | CLIP-IQA ↑ | DOVER ↑ |
|----------|---------|--------|---------|------------|---------|
| VideoLQ | DOVE-5B | 5.079 | 50.91 | 0.341 | 8.389 |
| | Ours-3B | 4.687 | 51.09 | 0.295 | 8.176 |
| AIGC28 | DOVE-5B | 4.486 | 62.62 | 0.610 | 16.13 |
| | Ours-3B | 3.801 | 62.99 | 0.561 | 15.77 |

benchmarks in Table 7. Our approach with 3B parameters shows superior performance on the metrics, including NIQE and MUSIQ, but is slightly inferior according to CLIP-IQA and DOVER. It is noticeable that DOVE has 5B parameters, i.e., about 1.67x larger than our 3B model. We further provide visual comparisons in Figure 9. Our 3B model is capable of generating more faithful details compared with DOVE.

## A.5 SPEED-QUALITY TRADE-OFF

We further provide metric performance of current multi-step diffusion-based restoration baselines under different sampling steps. As shown in Figure 10, with fewer sampling steps, the performance of most existing multi-step diffusion-based baselines (Zhou et al., 2024; Yang et al., 2024; He et al., 2024a; Xie et al., 2025) drops significantly in terms of perceptual metrics, including LPIPS, DISTS, MUSIQ, DOVER, and VMAF. Such speed-quality trade-off curves indicate that it is non-trivial to develop high-quality one-step diffusion-based methods for video restoration.

## A.6 COMPARISON WITH DLoRAL

We additionally compare with another concurrent work DLoRAL (Sun et al., 2025), as shown in Figure 11. While DLoRAL focuses on parameter-efficient finetuning a pretrained diffusion prior (Rombach et al., 2022) via LoRA (Hu et al., 2022b), it mostly relies on the frozen generative prior, which is not specifically designed for restoration tasks. Our work alternatively explores the potential improvement on real-world restoration brought by large-scale training with sufficient compute, which is underexplored by previous methods. The improvement brought by our proposed approach is evident in Figure 11, with superior textures of the sofa, skin of the woman, and details of the building and boat. The temporal consistency is also better with less texture flicking, as shown in the demo video of the supplementary material.

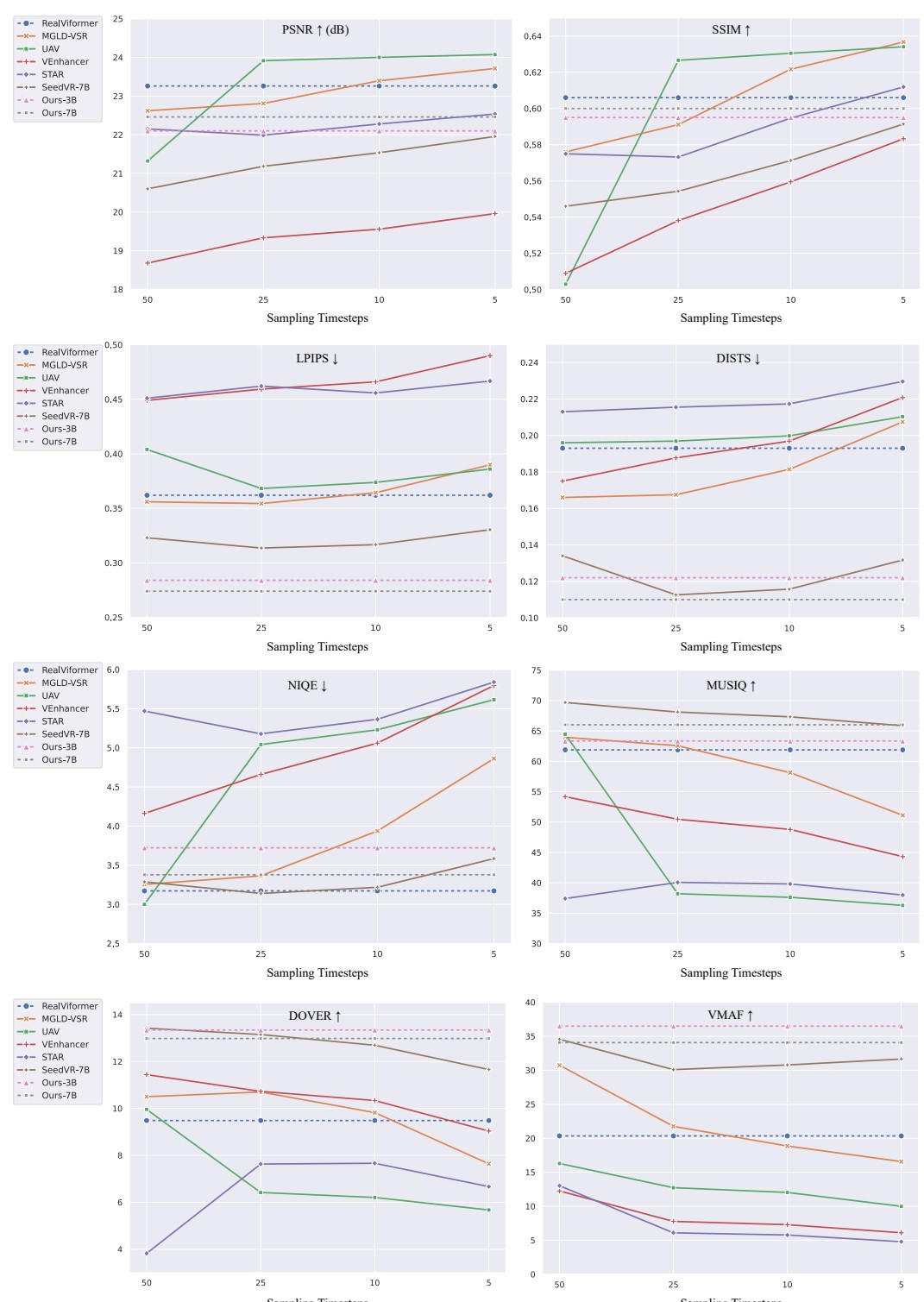

Figure 10: Quantitative comparison of speed-quality tradeoff (**Zoom-in for best view**).

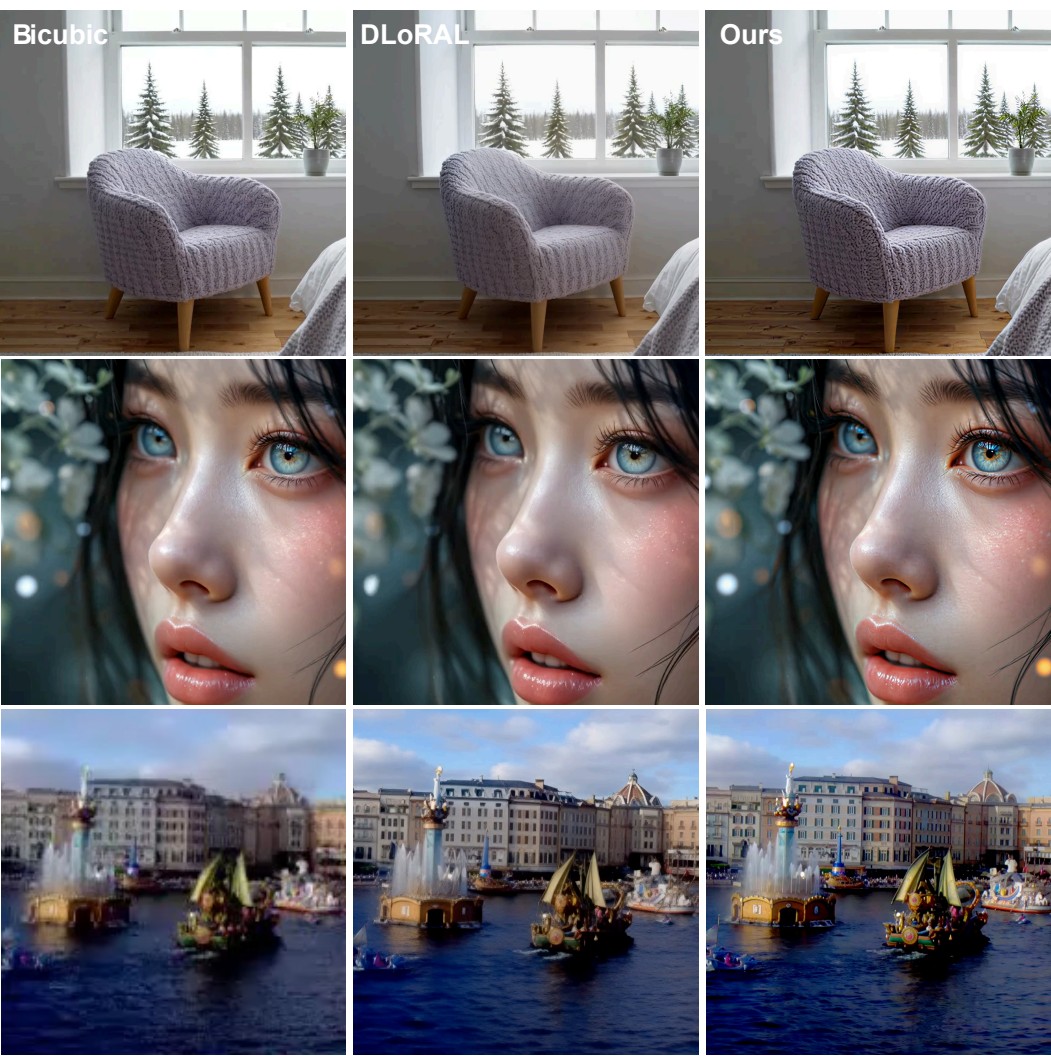

Figure 11: Qualitative comparison with DLoRAL (Sun et al., 2025) on real-world videos. Video comparison can be found in the demo video in the supplementary materials (**Zoom-in for best view**).

