# OpenReview forum: "SeedVR2: One-Step Video Restoration via Diffusion Adversarial Post-Training"
_ICLR.cc/2026/Conference — ICLR 2026 Poster_

### Official Review · Reviewer_BpPA · 2025-10-31

**Soundness:** 2
**Presentation:** 3
**Contribution:** 2
**Rating:** 4
**Confidence:** 5

**Summary:**

The paper proposes AnonymousVR, a one-step diffusion-based video restoration (VR) model that aims to overcome the high computational cost of traditional diffusion-based VR methods. The paper introduce an adaptive window attention mechanism that dynamically adjusts window sizes according to output resolution to address inconsistency issues in high-resolution VR. They also design an adversarial post-training strategy incorporating a feature matching loss to enhance stability and quality without significantly increasing training cost. Experimental results suggest that AnonymousVR achieves competitive or superior performance compared with existing multi-step VR methods while requiring only a single inference step.

**Strengths:**

The experimental results are promising, showing a substantial improvement.

**Weaknesses:**

1. The technical contributions of this paper appear rather weak, as the work seems to rely more on engineering efforts than on genuine technical innovation. The proposed window attention and adversarial diffusion training methods lack clear novelty and do not demonstrate substantial methodological advancement beyond existing approaches.

2. Please clarify the core improvements of the proposed adaptive window attention compared with prior window attention mechanisms, such as those in SeedVR. Additionally, please elaborate on the key advancements of your adversarial diffusion training strategy relative to existing adversarial diffusion methods.

3. The authors claim that their method is the first one-step video restoration approach; however, one-step super-resolution and restoration methods already exist in the literature (e.g., [1][2]). A more detailed discussion and comparison with these prior works are necessary to justify the claimed novelty.

[1] DOVE: Efficient One-Step Diffusion Model for Real-World Video Super-Resolution. NeurIPS 2025.

[2] One-Step Diffusion for Detail-Rich and Temporally Consistent Video Super-Resolution. NeurIPS 2025.

**Questions:**

see Weaknesses.

---

> ### Author Response · Authors · 2025-11-24
> **Response to Reviewer BpPA (1/2)**
>
> Dear Reviewer BpPA,
>
> We sincerely thank you for your positive feedback on **the promising experimental results with substantial improvement. We commit to making our code, models, and ComfyUI implementation publicly available to facilitate reproducibility and practical adoption of the community**. We address your questions below.
>
> **Q1. Technical contributions:**
> Thanks for the comments. We have provided a clear overview of the contributions in lines 88-112 of the paper. In one word, our paper tackles single-step video restoration (VR) with diffusion transformers in a large-scale adversarial training manner, an underexplored area where directly adopting existing components such as APT into VR is not effective and straightforward, as argued in our manuscript (Lines 88-112). Our core novelty lies in developing the crucial techniques that enable the successful training of a large-scale, one-step diffusion model for VR. These contributions include: 1) an adaptive window attention mechanism for multi-resolution inputs, 2) a tailored feature matching loss for perceptual quality, 3) a pre-distillation strategy for training stability, and 4) key observations, including the effectiveness of different loss functions for VR under large-scale training in Table 3. We would like to kindly remind the reviewer that the above practices have not been fully explored under the large-scale adversarial training manner by previous methods. Compared with previous VR methods, they typically rely on a pretrained generative prior like Stable Diffusion, which is already trained with large compute, but not specifically designed for the restoration task. Our method presents an alternative successful practice to train a powerful one-step diffusion model and verifies the effectiveness with extensive experiments.
>
> The value of these contributions was affirmed by other reviewers. They specifically highlighted the adaptive attention as a new setting (Reviewer WNGK) and "key contribution" (Reviewer Q1GV), the feature matching loss as another "key contribution" (Reviewer Q1GV) and "a reasonable, efficiency-motivated tweak" (Reviewer WNGK), and appreciated our overall approach for its "meaningful step beyond prior one-step image restoration" (Reviewer WNGK). This feedback from both experts reinforces our belief that our work offers a novel and valuable contribution to the field.
>
>
> **Q2. Clarification on core contributions:**
> Thanks for the comments. We have provided an overview of the motivations and significance of our contributions, including the proposed adaptive window attention and our adversarial diffusion training strategy in Lines 88-112 of our paper. Specifically, previous related methods, such as SeedVR, typically rely on window attention with a fixed window size, limiting inference robustness with potential artifacts for high-resolution restoration, which is a crucial problem for video restoration. We significantly enhance the robustness under such scenarios with the proposed adaptive window attention, enabling dynamic adjustment of the window size within a certain range, as evident in Figure 4 of the paper.
>
> For our training strategy, we present a successful practice on training a large-scale adversarial restoration model, which has not been explored in previous restoration methods. Key advancements include 1) the thorough analysis of effect of different loss combinations for adversarial training specific to the restoration task in Table 3 of the paper; 2) the feature matching loss to replace LPIPS for much more efficient adversarial training, which is ''a reasonable, efficiency-motivated tweak (not conceptually new, but well-justified here)'' (Reviewer WNGK), and 3) the verification of the effectiveness of the pre-distillation strategy specific to the restoration task.

---

> ### Author Response · Authors · 2025-11-24
> **Response to Reviewer BpPA (2/2)**
>
> **Q3. Comparison with concurrent work:**
> Thanks for the suggestions. Due to the close date between the opensource of NeurIPS and the submission of ICLR, DOVE and DLoraL should be considered as concurrent work with ours. To avoid misunderstanding, we have modified the claim from ''among the first'' to ''among the early'' in the revision. We have provided quantitative and qualitative comparisons with DOVE in A.4 of the appendix. We further follow the suggestion to add the comparison with DLoraL in A.7 of the revision. The improvement brought by our proposed approach is evident in Figure 10, with superior textures of the sofa, skin of the woman, and details of the building and boat. The temporal consistency is also better with less texture flicking, as shown in the demo video "`2652_demo_rebuttal.mp4`" of the supplementary material.
>
> While DOVE and DLoraL focus on finetuning a pre-trained generative prior such as CogVideoX [Ref. A] and Stable Diffusion for one-step restoration, either by small-scale finetuning or LoRA, they largely rely on almost the same network architecture and training scheme as the generative prior, which are not specifically designed for the restoration task. Unlike these concurrent methods, our approach explores an alternative way by tackling single-step video restoration (VR) with diffusion transformers in a large-scale adversarial training manner, with specific designs and verifications towards the restoration task, including efficient and effective attention and loss designs, as well as training strategies. These discussions are included in the revision.
>
>
> [Ref. A] Zhuoyi Yang, et al. CogVideoX: Text-to-Video Diffusion Models with An Expert Transformer. ICLR, 2025.

---

> > ### Comment · Reviewer_BpPA · 2025-11-28
> >
> > Thank you very much for the clarification on the novelty and the additional experiments, which have addressed most of my concerns. I have decided to raise my score to 6. Since OpenReview currently does not allow score updates, I will update it once the system permits, or I will inform the AC directly.

---

> > > ### Author Response · Authors · 2025-11-28
> > >
> > > Dear Reviewer BpPA,
> > >
> > > Thanks for your comments. We would like to let you know that we always appreciate and respect your professional comments on helping us improve the paper quality. This will not be affected by any non-academic factors.

---

### Official Review · Reviewer_Q1GV · 2025-10-31

**Soundness:** 3
**Presentation:** 3
**Contribution:** 3
**Rating:** 6
**Confidence:** 4

**Summary:**

This paper proposes a one-step video restoration (VR) model that leverages a diffusion adversarial post-training (APT) framework to perform high-resolution video restoration. The key contributions include an adaptive window attention mechanism for high-resolution inputs, and a robust feature matching loss for improved training stability. The model is evaluated against state-of-the-art VR methods, demonstrating competitive performance while being four times faster than existing diffusion-based VR methods. The results suggest that it achieves comparable or even superior performance in real-world scenarios, especially with AI-generated content and high-resolution videos.

**Strengths:**

1. The paper introduces a novel one-step VR method by applying APT to diffusion-based models, reducing the computational burden significantly compared to traditional multi-step approaches.

2. The adaptive window attention mechanism for handling high-resolution videos and the feature matching loss for training stability are key contributions that improve the model's performance and robustness across varying video resolutions.

3. The method shows promising quantitative and qualitative results, outperforming existing VR approaches in real-world and synthetic benchmarks, demonstrating significant gains in speed and restoration quality.

**Weaknesses:**

1. The paper lacks comparisons with the latest VSR methods presented at NeurIPS 2025 (such as DLoraL [1] and DOVE [2]). The authors should include comparisons with these methods to better demonstrate the competitiveness of the proposed approach.

2. The paper does not provide results trained on public datasets (such as REDS). The reported improvements might stem from using a larger private dataset. Will the authors make the dataset publicly available?

3. Despite achieving faster inference, the training requires 72 H100 GPUs and significant resources, which raises concerns about scalability and accessibility for broader research adoption.

[1] One-Step Diffusion for Detail-Rich and Temporally Consistent Video Super-Resolution. NeurIPS2025

[2] DOVE: Efficient One-Step Diffusion Model for Real-World Video Super-Resolution. NeurIPS2025

**Questions:**

See weaknesses.

---

> ### Author Response · Authors · 2025-11-24
> **Response to Reviewer Q1GV**
>
> Dear Reviewer Q1GV,
>
> We sincerely thank you for your positive feedback on **the novelty and efficiency of our approach, key contributions of adaptive window attention and feature matching loss, and promising results. We commit to making our code, models, and ComfyUI implementation publicly available to facilitate reproducibility and practical adoption of the community**. We address your questions below.
>
> **Q1. Comparison with concurrent works:**
> Thanks for the suggestions. Due to the close date between the opensource of NeurIPS and the submission of ICLR, DOVE and DLoraL should be considered as concurrent work with ours. We have provided quantitative and qualitative comparisons with DOVE in A.4 of the appendix. We further follow the suggestion to add the comparison with DLoraL in A.7 of the revision. While DLoRAL focuses on parameter-efficient finetuning a pretrained diffusion prior (which is also large-scale trained), it may share similar generative bias with the frozen generative prior. Our work alternatively explores the potential improvement on real-world restoration brought by large-scale training with sufficient compute, which is underexplored by previous methods. The improvement brought by our proposed approach is evident in Figure 10, with superior textures of the sofa, skin of the woman, and details of the building and boat. The temporal consistency is also better with less texture flicking, as shown in the demo video "`2652_demo_rebuttal.mp4`" of the supplementary material.
>
> **Q2. Training data:**
> Thanks for the comments. Existing diffusion-based methods like UAV, MGLD-VSR, DloraL and DOVE generally rely on pretrained diffusion prior such as Stable Diffusion, Stable Diffusion ×4 upscaler and CogVideoX [Ref. A]. It is notable that these priors are also trained on large-scale datasets, some of which are private. Similar to these generation-based models, some of our training data cannot be made public due to copyright constraints. However, we believe our model could serve as a strong prior for future works to eliminate their need of large-scale data training for restoration.
>
> **Q3. Training compute:**
> Thanks for the comments. The motivation behind such compute is that we would like to explore the potential improvement on real-world restoration brought by scaling up training with sufficient compute, which is underexplored by previous methods. With limited compute, previous methods typically rely on a pretrained generative prior like Stable Diffusion, which is already trained with large compute, but not specifically designed for the restoration task. We instead present an alternative successful practice to train a powerful one-step diffusion model via proposing efficient and effective attention and loss designs, as well as training strategies. When considering the training cost of the generative prior used in previous diffusion-based methods, the compute gap between our approach and these methods should be acceptable from our view. Moreover, given recent advances of large-scale training in image and video generation [Ref. B, Ref. C, Ref. D, Ref. E] and image restoration [Ref. F], we believe our work provides a timely exploration towards video restoration with one-step sampling. Finally, with code release, we believe our model could be a superior generative prior compared with existing ones to help typical academic labs develop their own restoration approaches with state-of-the-art performance. Moreover, equipped with popular optimizations used in ComfyUI, our model supports inference on a consumer GPU with a VRAM of 12GB or even less, making it possible for practical accessibility. We will release the implementations for community use later.
>
> [Ref. A] Zhuoyi Yang, et al. CogVideoX: Text-to-Video Diffusion Models with An Expert Transformer. ICLR, 2025.
> [Ref. B] Sauer, Axel, et al. "Fast high-resolution image synthesis with latent adversarial diffusion distillation." SIGGRAPH Asia, 2024.
> [Ref. C] Esser, Patrick, et al. "Scaling rectified flow transformers for high-resolution image synthesis." ICML, 2024.
> [Ref. D] Yang, Zhuoyi, et al. "CogVideoX: Text-to-Video Diffusion Models with An Expert Transformer." ICLR, 2025.
> [Ref. E] Kondratyuk, Dan, et al. "VideoPoet: a large language model for zero-shot video generation." ICML, 2024.
> [Ref. F] Yu, Fanghua, et al. "Scaling up to excellence: Practicing model scaling for photo-realistic image restoration in the wild." CVPR, 2024.

---

> ### Comment · Reviewer_Q1GV · 2025-11-28
>
> Thank you to the authors for the detailed rebuttal. The authors' responses have addressed my main concerns, and I am maintaining my positive rating.

---

> > ### Author Response · Authors · 2025-11-28
> >
> > Dear Reviewer Q1GV,
> >
> > Thanks for your comments. We would like to let you know that we always appreciate and respect your professional comments on helping us improve the paper quality. This will not be affected by any non-academic factors.

---

### Official Review · Reviewer_rqaT · 2025-10-31

**Soundness:** 3
**Presentation:** 3
**Contribution:** 2
**Rating:** 2
**Confidence:** 5

**Summary:**

The paper proposes AnonymousVR, a one-step diffusion-based video restoration method trained with adversarial post-training. It starts from a strong diffusion transformer, then applies progressive distillation and full adversarial tuning to remove the multi-step sampling cost. In addition, an adaptive window attention that adjusts window size to input resolution to avoid block boundaries at high resolutions, and a feature-matching loss taken from discriminator layers to replace expensive LPIPS during training are proposed. Experiments on synthetic, real, and AIGC videos demonstrate the effectiveness of the proposed method.

**Strengths:**

- The introduction of adaptive window attention effectively reduces boundary artifacts when processing high-resolution frames.

- The training strategy which combines RpGAN, approximate R2 regularization, feature-matching losses, and progressive distillation to ensure stable convergence and high perceptual quality is comprehensive.

- The experiments are extensive and include both synthetic and real-world data, multiple objective and perceptual metrics, as well as a well-organized user study.

**Weaknesses:**

- My main concern is that the novelty of the method is somewhat limited, as it largely builds upon the existing Adversarial Post-Training (APT) framework, and the paper does not clearly explain the fundamental differences or new contributions beyond APT.

- The training process is extremely resource-intensive, requiring 72 H100 GPUs, which significantly limits reproducibility and practical accessibility.

- The method’s robustness under challenging conditions, such as heavy degradations, large motion, or complex real-world dynamics, appears limited.

**Questions:**

See weaknesses.

---

> ### Author Response · Authors · 2025-11-24
> **Response to Reviewer rqaT (1/2)**
>
> Dear Reviewer rqaT,
>
> We sincerely thank you for your positive feedback on **the effectiveness of our proposed adaptive window strategy, comprehensive training strategies for ensuring stable convergence and high perceptual quality, and extensive experiments. We commit to making our code, models, and ComfyUI implementation publicly available to facilitate reproducibility and practical adoption of the community**. We address your questions below.
>
>
> **Q1. Novelty:**
> Thanks for the comments. We have provided a clear overview of the contributions as well as the differences beyond APT in lines 88-112 of the paper. In one word, our paper tackles single-step video restoration (VR) with diffusion transformers in a large-scale adversarial training manner, an underexplored area where directly adopting existing components such as APT into VR is not effective and straightforward, as argued in our manuscript (Lines 88-112). Our core novelty lies in developing the crucial techniques that enable the successful training of a large-scale, one-step diffusion model for VR. These contributions include: 1) an adaptive window attention mechanism for multi-resolution inputs, 2) a tailored feature matching loss for perceptual quality, 3) a pre-distillation strategy for training stability, and 4) key observations, including the effectiveness of different loss functions for VR under large-scale training in Table 3.
>
> The value of these contributions was affirmed by other reviewers. They specifically highlighted the adaptive attention as a new setting (Reviewer WNGK) and "key contribution" (Reviewer Q1GV), the feature matching loss as another "key contribution" (Reviewer Q1GV) and "a reasonable, efficiency-motivated tweak" (Reviewer WNGK), and appreciated our overall approach for its "meaningful step beyond prior one-step image restoration" (Reviewer WNGK). This feedback from both experts reinforces our belief that our work offers a novel and valuable contribution to the field.
>
> **Q2. Training compute:**
> Thanks for the comments. The motivation behind such compute is that we would like to explore the potential improvement on real-world restoration brought by large-scale training with sufficient compute, which is underexplored by previous methods. With limited compute, previous methods typically rely on a pretrained generative prior like Stable Diffusion, which is already trained with large compute, but not specifically designed for the restoration task. We instead present an alternative successful practice to train a powerful one-step diffusion model via proposing efficient and effective attention and loss designs, as well as training strategies. When taking into account the training cost of the generative prior used in previous diffusion-based methods, the compute gap between our approach and these methods should be acceptable from our view. Moreover, given recent advances of large-scale training in image and video generation [Ref. A, Ref. B, Ref. C, Ref. D] and image restoration [Ref. E], we believe our work provides a timely exploration towards video restoration with one-step sampling. Finally, with code release, we believe our model could be a superior generative prior compared with existing ones to help typical academic labs develop their own restoration approaches with state-of-the-art performance. Moreover, equipped with popular optimizations used in ComfyUI, our model supports inference on a consumer GPU with a VRAM of 12GB or even less, making it possible for practical accessibility. We will release the implementations for community use later.
>
> Although the training process is resource-intensive, our inference is highly efficient thanks to our proposed one-step diffusion framework and window-based attention mechanism. As shown in Fig. 1 of the main paper, our model achieves a 10× reduction in FLOPs and is 4× faster than diffusion-based baselines during inference. With this strong performance and high efficiency, we believe our approach offers substantial potential for future research in real-world video restoration.
>
> **Q3. Robustness under challenging conditions:**
> Thanks for the comments. Restoring challenging conditions such as heavy degradations, large motion, or complex real-world dynamics has been a long-standing problem for real-world video restoration. While our method cannot always achieve perfect performance under such cases (as discussed in A.5 of the appendix), we argue that our method is capable of achieving state-of-the-art performance, as shown in A.8 of the revision and the newly provided demo video "`2652_demo_rebuttal.mp4`" in the supplementary material.

---

> ### Author Response · Authors · 2025-11-24
> **Response to Reviewer rqaT (2/2)**
>
> [Ref. A] Sauer, Axel, et al. "Fast high-resolution image synthesis with latent adversarial diffusion distillation." SIGGRAPH Asia, 2024.
> [Ref. B] Esser, Patrick, et al. "Scaling rectified flow transformers for high-resolution image synthesis." ICML, 2024.
> [Ref. C] Yang, Zhuoyi, et al. "CogVideoX: Text-to-Video Diffusion Models with An Expert Transformer." ICLR, 2025.
> [Ref. D] Kondratyuk, Dan, et al. "VideoPoet: a large language model for zero-shot video generation." ICML, 2024.
> [Ref. E] Yu, Fanghua, et al. "Scaling up to excellence: Practicing model scaling for photo-realistic image restoration in the wild." CVPR, 2024.

---

> > ### Comment · Reviewer_rqaT · 2025-11-27
> > **good rebuttal**
> >
> > After carefully reading the other reviewers’ feedback and the authors’ rebuttal, I find that my primary concerns regarding the novelty and robustness under challenging conditions have been satisfactorily resolved. I am therefore raising my score accordingly.

---

> > > ### Author Response · Authors · 2025-11-28
> > >
> > > Dear Reviewer rqaT,
> > >
> > > Thanks for your comments. We would like to let you know that we always appreciate and respect your professional comments on helping us improve the paper quality. This will not be affected by any non-academic factors.

---

### Official Review · Reviewer_WNGK · 2025-11-01

**Soundness:** 3
**Presentation:** 3
**Contribution:** 3
**Rating:** 8
**Confidence:** 4

**Summary:**

I think the paper tackles an important and timely problem: high-resolution one-step video restoration (VR). The method “AnonymousVR” initializes from a diffusion transformer (SeedVR) and then performs adversarial post-training (APT) to convert it into a single-step generator. The paper’s two main technical levers are:

(1) an adaptive window attention to avoid high-res window boundary artifacts.

(2) an adversarial post-training recipe with progressive distillation and a feature-matching loss (taken from the discriminator) to stabilize training while avoiding pixel-space LPIPS cost. Experiments suggest competitive or better perceptual quality vs multi-step diffusion VR at much lower latency.

**Strengths:**

- I think the jump to truly one-step VR with a diffusion transformer (initialized from SeedVR) plus APT is a meaningful step beyond prior one-step image restoration; prior works are mostly teacher-distillation or rely on fixed diffusion priors that cap quality. This work claims distillation-free adversarial post-training against real data after a lightweight progressive distillation stage to bridge the gap, which is interesting for video.

- The adaptive window attention to handle arbitrary resolutions with dynamic window size feels practical and addresses a real artifact at 2K/1080p; to my knowledge, such resolution-consistent windowing for VR in a one-step setting is new.

- Using the discriminator’s multi-layer features as an LPIPS surrogate in latent / discriminator space for high-res VR is a reasonable, efficiency-motivated tweak (not conceptually new, but well-justified here).

Overall, I think the contribution is incremental-to-moderate in theory but practically impactful for high-res, fast VR.

**Weaknesses:**

- I am concerned about the compute-heaviness. I think the approach relies heavily on significant compute (72×H100, 10M/5M pairs), which limits reproducibility in typical academic labs despite code release plans. Claims of “largest-ever VR GAN” underscore this.

- Scope of degradations. While synthetic degradations follow prior work, I think the paper could better characterize real-world degradation diversity and robustness (e.g., compression artifacts, rolling shutter, severe motion blur) beyond VideoLQ/AIGC28; the method’s failure cases are not deeply analyzed.

- Fairness of baselines. Diffusion baselines are run with 50 steps “to maintain stable performance”; I think it would be fair to include their fastest-setting curves (e.g., 10/25/50 steps trade-off plots) to contextualize speed-quality trade-offs.

- Temporal metrics and consistency are missing. The paper mostly emphasizes frame-wise perceptual metrics; I would expect temporal consistency metrics (e.g., t-LPIPS variants or VMAF-like temporal terms) or user study questions specific to flicker/temporal stability. Current user study aggregates “overall quality” but not explicitly “temporal coherence.”

**Questions:**

- Temporal stability: How does one-step AnonymousVR compare to SeedVR on temporal coherence (quantitative & qualitative)? Any metric beyond a user study that isolates flicker?

- Feature-matching loss: Why pick discriminator blocks 16/26/36 specifically? Did you try earlier/later layers or a learned weighting per layer? Impact on training speed?

- Progressive distillation details: What are the teacher/student schedules and hyper-params across strides (64→32→…→1)? How much of the final gain comes from progressive distillation vs APT?

---

> ### Author Response · Authors · 2025-11-24
> **Response to Reviewer WNGK (1/2)**
>
> Dear Reviewer WNGK,
>
> We sincerely appreciate your positive feedback on the **meaningfulness of our work, novelty of dynamic window attention, and the significance of the feature loss. We commit to making our code, models, and ComfyUI implementation publicly available to facilitate reproducibility and practical adoption of the community.** We address your questions below (Q1-Q4 for the Weakness part, and Q4-Q6 for the Question part).
>
> **Q1. Computational cost:**
> Thanks for the comments. The motivation behind such compute is that we would like to explore the potential improvement on real-world restoration brought by large-scale training with sufficient compute, which is underexplored by previous methods. With limited compute, previous methods typically rely on a pretrained generative prior like Stable Diffusion, which is already trained with large compute, but not specifically designed for the restoration task. We instead present an alternative successful practice to train a powerful one-step diffusion model via proposing efficient and effective attention and loss designs, as well as training strategies. Regarding the training cost of the generative prior used in previous diffusion-based methods, the compute gap between our approach and these methods should be acceptable from our view. Moreover, given recent advances in large-scale training in image and video generation [Ref. A, Ref. B, Ref. C, Ref. D] and image restoration [Ref. E], we believe our work provides a timely exploration towards video restoration with one-step sampling. Finally, with code release, we hope our model could be a superior generative prior compared with existing ones to help typical academic labs develop their own restoration approaches with SOTA performance.
>
> **Q2. Scope of degradations:**
> Thanks for the comments. VideoLQ contains real-world videos with different resolutions and contents to cover as many degradations as possible [Ref. F], which should cover real-world degradations such as compression artifacts and severe motion blur. We agree that collecting a larger-scale real-world benchmark containing diverse real-world degradations could be a promising future work. We have discussed the limitations of our method, including failure cases in Sec. A.5 of Appendix in the paper.
>
> **Q3. Fairness of baselines:**
> Thanks for the comments. We follow the suggestion to provide the trade-off plot of diffusion-based baselines, including MGLD-VSR, UAV, VEnhancer, STAR and SeedVR-7B in Figure 9 in the revision across multiple commonly used metrics. The superiority of our method in terms of various perceptual metrics, including LPIPS, DISTS, MUSIQ, DOVER and VMAF, becomes more evident compared with these baselines in fewer sampling steps.
>
>
> **Q4. Temporal metrics and stability:**
> Thanks for the comments. We have presented metrics that consider temporal consistency in our paper, e.g., DOVER in Table 1 and VMAF in Table 4 of the paper, where our methods achieve comparable or superior performance compared with existing baselines. As for t-LPIPS, if our understanding is correct, such a metric is used by calculating the difference between two adjacent frames in feature space. We think t-LPIPS may favor oversmooth videos and penalize texture generation, since without a ground-truth reference, oversmooth results generally lead to fewer differences across frames, which is not preferred by human subjects. While temporal consistency is not explicitly measured in our user study, it plays a key role in the 'overall quality', which we also stess the focus on the temporal consistency in our user study guidance. We will make this clear in the paper.

---

> ### Author Response · Authors · 2025-11-24
> **Response to Reviewer WNGK (2/2)**
>
> **Q5. Feature-matching Loss:**
> Thanks for the comments. We follow the discriminator design of APT and intuitively extract the features before the attention-only transformer blocks (i.e., 16/26/36 blocks), where features are closely related to the multi-scale predictions of the discriminator for loss calculation. Based on our understanding, the impact on training speed with different feature extraction strategies should be negligible compared with the naive LPIPS loss, where a heavy decoder is required to decode latent features into the pixel space.
>
> **Q6. Progressive distillation details:**
> Thanks for the comments. During the distillation, loss is calculated on the vector field following Flow matching [Ref. G]. Both teacher and student models adopt the linear noise schedule with a timestep between 0 and 999. The teacher model uses Euler sampler during training with a CFG scale of 7.5 for 64 timesteps and 1.0 for others. We adopt AdamW [Ref. H] with a weight decay of 0.01 as optimizer and the learning rate is set to 1e-6.
>
> We have verified the gain of progressive distillation in the last column in Table 3 and the effectiveness of APT in Table 5.
>
> [Ref. A] Sauer, Axel, et al. "Fast high-resolution image synthesis with latent adversarial diffusion distillation." SIGGRAPH Asia, 2024.
> [Ref. B] Esser, Patrick, et al. "Scaling rectified flow transformers for high-resolution image synthesis." ICML, 2024.
> [Ref. C] Yang, Zhuoyi, et al. "CogVideoX: Text-to-Video Diffusion Models with An Expert Transformer." ICLR, 2025.
> [Ref. D] Kondratyuk, Dan, et al. "VideoPoet: a large language model for zero-shot video generation." ICML, 2024.
> [Ref. E] Yu, Fanghua, et al. "Scaling up to excellence: Practicing model scaling for photo-realistic image restoration in the wild." CVPR, 2024.
> [Ref. F] Kelvin C.K. Chan, et al. Investigating Tradeoffs in Real-World Video Super-Resolution. CVPR, 2022.
> [Ref. G] Yaron Lipman, et al. Flow Matching For Generative Modeling. ICLR, 2023.
> [Ref. H] Adam, Kingma DP Ba J. A method for stochastic optimization. ArXiv, 2014.

---

> > ### Comment · Reviewer_WNGK · 2025-11-27
> >
> > Thank you for the detailed rebuttal and additional analyses.
> >
> > Your clarification on the motivation for large-scale training and the role of your model as a reusable restoration prior alleviates most of my concerns about compute, especially with the commitment to release code/models/ComfyUI implementations. The added trade-off plots for diffusion baselines across different sampling steps address my fairness concerns and make the speed–quality story much clearer.
> >
> > Overall, the rebuttal satisfactorily addresses my main questions and strengthens the paper. I maintain my positive assessment and keep my score at 8 (accept, good paper).

---

> > > ### Author Response · Authors · 2025-11-27
> > >
> > > Dear Reviewer WNGK,
> > >
> > > Thanks for your constructive feedback to help strengthen our paper. We sincerely appreciate your valuable comments and suggestions!

---

### Author Response · Authors · 2025-11-24
**Response to All Reviewers and AC**

Dear AC and Reviewers,

We sincerely thank you for your time and valuable comments during the review and discussion process. We have responded individually to each reviewer to address any concerns. Below, we provide a brief overview of our work and its contributions.

As an early attempt towards single-step video restoration (VR) with diffusion transformers, our proposed method demonstrates strong performance and efficiency compared to prior works. Given recent advances of **large-scale training** in image and video generation [Ref. A, Ref. B, Ref. C, Ref. D] and image restoration [Ref. E], it is non-trivial to investigate its successful practices in video restoration. Our paper tackles single-step video restoration (VR) with diffusion transformers in a large-scale adversarial training manner, which is still underexplored. Our **core contributions** lie in developing the crucial techniques that enable the successful training of a large-scale, one-step diffusion model for VR, including: 1) an adaptive window attention mechanism for multi-resolution inputs, 2) a tailored feature matching loss for perceptual quality, 3) a pre-distillation strategy for training stability, and 4) key observations, including the effectiveness of different loss functions for VR under large-scale training in Table 3.


We appreciate the recognition of the reviewers from the following aspects: **a) Novel Methodology**: meaningful step beyond prior one-step restoration via distillation-free APT (R-WNGK), novel application of APT to diffusion models (R-Q1GV), adaptive window attention effectively handles arbitrary resolutions and boundary artifacts (R-WNGK, R-rqaT, R-Q1GV). **b) Stable Training**: comprehensive training strategy ensuring stable convergence (R-rqaT), well-justified feature matching loss for stability (R-Q1GV, R-WNGK). **c) Superior Performance**: extensive experiments on synthetic and real-world data (R-rqaT), significant gains in speed and restoration quality (R-Q1GV), substantial improvement over existing approaches (R-BpPA).


We believe our work can serve as a strong baseline, inspiring future research to adopt large-scale adversarial training for video enhancement and beyond. **We will open-source our code and model to ensure reproducibility and to support further development by the community.** With specific designs towards the restoration task, we hope our model could be a superior generative prior compared with existing ones to help typical academic labs develop their own restoration approaches with state-of-the-art performance.

Finally, we sincerely appreciate the reviewers’ constructive feedback. **According to the reviewers' comments, we have improved our revision with additional experimental results and provided an additional video demo in our supplementary materials.** We also thank the AC for the time of handling the review of our paper.

We further make a **summary of the feedback** from reviewers:
- R-WNGK: The reviewer highlights that our clarification of the model as a "reusable restoration prior", combined with the commitment to release code/models (including ComfyUI), effectively justified the large-scale training motivation; our new trade-off plots (sampling steps vs. quality) for diffusion baselines clarified the speed-quality advantage, making the comparison fair and convincing. Overall, the reviewer is satisfied with our response and promise to maintain his positive assessment.
- R-rqaT: The reviewer regards our response as a "good rebuttal", and indicates that his primary concerns regarding the novelty and robustness under challenging conditions have been satisfactorily resolved. The reviewer promises to raise the score to a positive rating (He made it before the score reverted).
- R-Q1GV: The reviewer claims that his concerns have been addressed and will maintain his positive rating.
- R-BpPA : The reviewer is satisfied with our response to the novelty concerns and additional experimental results regarding the comparison with concurrent works. The reviewer promises to raise the score to a positive rating.

Thanks,
Authors

[Ref. A] Sauer, Axel, et al. "Fast high-resolution image synthesis with latent adversarial diffusion distillation." SIGGRAPH Asia, 2024.
[Ref. B] Esser, Patrick, et al. "Scaling rectified flow transformers for high-resolution image synthesis." ICML, 2024.
[Ref. C] Yang, Zhuoyi, et al. "CogVideoX: Text-to-Video Diffusion Models with An Expert Transformer." ICLR, 2025.
[Ref. D] Kondratyuk, Dan, et al. "VideoPoet: a large language model for zero-shot video generation." ICML, 2024.
[Ref. E] Yu, Fanghua, et al. "Scaling up to excellence: Practicing model scaling for photo-realistic image restoration in the wild." CVPR, 2024.

---

### Meta-Review · Area_Chair_CwWZ · 2025-12-23

**Summary:**

Reviewers agreed the paper tackles a timely problem—one-step, high-resolution video restoration with diffusion transformers—and found the approach practically impactful. The strengths repeatedly noted include adaptive window attention for arbitrary-resolution inference, a feature-matching loss that improves stability/efficiency versus pixel-space perceptual losses, and extensive experiments showing strong speed–quality trade-offs. The key concerns shaping the discussion were (i) novelty beyond existing APT-style frameworks, (ii) very heavy training compute and reproducibility/accessibility, and (iii) fairness/completeness of comparisons, including fast-step diffusion baselines, temporal consistency evaluation, and concurrent one-step VSR methods.

**Reviewer Concerns:**

Addressed by rebuttal:

Added speed–quality trade-off plots across sampling steps, which clarified the narrative and reduced the “50-step only” concern.

Added comparisons with DOVE / DLoraL and clarified “early attempt” positioning.

Explicitly articulated what is new beyond APT when moving to large-scale one-step VR with diffusion transformers.

Strengthened empirical support with DOVER/VMAF and additional demo material.

Still outstanding:

Training compute remains extremely large (72×H100), so reproducibility is still a real barrier even with code/model release.

Training data cannot be fully released, leaving some uncertainty about how much gains rely on private-scale data.

Temporal consistency evaluation is improved but still could be more direct/diagnostic, though existing evidence is reasonably supportive.

**Reviewer Scores:**

R-WNGK: 8 → 8

R-rqaT: 2 → 6

R-Q1GV: 6 → 6

R-BpPA: 4 → 6

---

### Decision · Program_Chairs · 2026-01-26

Accept (Poster)